# CLASS-CONDITIONAL CONFORMAL PREDICTION FOR IMBALANCED DATA VIA TOP-$k$ CLASSES

## ABSTRACT

Classification tasks where data contains skewed class proportions (aka *imbalanced data*) arises in many real-world applications including medical diagnosis. Safe deployment of classifiers for imbalanced data settings require theoretically-sound uncertainty quantification. Conformal prediction (CP) is a promising framework for producing prediction sets from black-box classifiers with a user-specified coverage (i.e., true class is contained with high probability). Existing class-conditional CP (CCP) method employs a black-box classifier to find one threshold for each class during calibration and then includes every class label that meets the corresponding threshold for testing inputs, leading to large prediction sets. This paper studies the problem of how to develop provable CP methods with small prediction sets for the class-conditional coverage setting and makes several contributions. First, we theoretically show that marginal CP can perform arbitrarily poorly and cannot provide coverage guarantee for minority classes. Second, we propose a principled algorithm referred to as $k$-*Class-conditional CP ($k$-CCP)*. The $k$-CCP method estimates class-specific non-conformity score threshold using inflated coverage and calibrated rank threshold depending on the top-$k$ error of the classifier for each class. Given a testing input, $k$-CCP includes only those class labels which satisfy both class-specific thresholds for score and calibrated rank to produce the prediction set. We prove that $k$-CCP provides class-conditional coverage and produces smaller prediction sets over the CCP method. Our experiments on benchmark datasets demonstrate that $k$-CCP achieves class-conditional coverage and produces significantly smaller prediction sets over baseline methods.

## 1 INTRODUCTION

Many real-world applications such as those in healthcare domain inherently exhibit classification data with long-tailed distributions of class labels, i.e., *imbalanced data* (He and Garcia, 2009). The labels with high and low frequency are referred to as *majority* and *minority* classes respectively. In imbalanced data settings, the minority classes are very important (e.g., cancer class in medical diagnosis) (Mazurowski *et al.*, 2008). For safe deployment of ML classifiers in such imbalanced data scenarios, we require uncertainty quantification (UQ) methods with strong theoretical guarantees.

Conformal prediction (CP) (Shafer and Vovk, 2008) captures the deviation of the predicted label from the true label in the form of a *prediction set* (subset of candidate class labels). For example, in medical diagnosis applications, such prediction sets will allow the doctor to rule out lung cancer even though the most likely diagnosis is flu. CP provides theoretical guarantees for a given target coverage: true label is present in the prediction set with a user-specified probability 1 - $\alpha$ (e.g., 90%). CP methods involve two main steps: 1) Employ a trained classifier (e.g., neural network) to compute *conformity scores* which measure the similarity between calibration data and a testing input; and 2) Using the conformity scores on calibration data to find a threshold for producing prediction sets to satisfy the marginal coverage constraint 1 - $\alpha$. It is easy to achieve high coverage by producing large prediction sets (e.g., all candidate labels in the worst-case), but it increases the burden of human expert in human-ML collaborative systems (Straitouri *et al.*, 2023; Babbar *et al.*, 2022). Therefore, CP methods which produce smaller prediction sets by satisfying the target coverage are desirable in practice. The main research question of this paper is: *how can we develop provable CP methods for the imbalanced data setting to produce small prediction sets?* To answer the question, this paper proposes a novel algorithm referred to as $k$-*Class-Conditional Conformal Prediction ($k$-CCP)*.

In spite of the recent successes of CP (Angelopoulos and Bates, 2021), there is relatively less work for imbalanced datasets to rigorously understand the challenges for CP. We first derive class-conditional coverage bounds for marginal CP and demonstrate that without further assumptions, marginal CP can perform arbitrarily poor on some classes (e.g., minority ones). The key idea behind $k$-CCP is to use the rank order of the candidate class labels from the classifier to modify the calibration procedure and the mechanism to produce prediction sets. As the name suggests, $k$-CCP uses a single conformity scoring function based on the trained classifier and performs calibration for each class label separately. For each class, $k$-CCP estimates the conformity score threshold using inflated coverage and the calibrated rank threshold depending on the top-$k$ accuracy of the classifier for that class. Given a testing input, $k$-CCP includes only those labels which satisfy both class-specific thresholds for score and calibrated rank to produce the prediction set.

The primary CP method for imbalanced data (Vovk, 2012; Sadinle *et al.*, 2019; Angelopoulos and Bates, 2021) which we refer to as CCP differs from $k$-CCP in two ways. First, CCP only estimates class-specific conformity score threshold to achieve $1 - \alpha$ coverage for every class. $k$-CCP performs double-calibration: one for score threshold (same as CCP) and one for calibrated rank threshold for each of the classes. It uses different inflated coverage for each class $c$: $1 - \tilde{\alpha}_c > 1 - \alpha$ (e.g., 91% instead of 90%). The inflated coverage is used to enable the second condition based on calibrated rank thresholds for different classes to achieve improved trade-off between class-conditional coverage and prediction set size. Second, to produce prediction sets for every testing input, CCP iterates over all classes by comparing the conformity score of each class with the corresponding threshold. For each class label, $k$-CCP compares with the corresponding conformity score and calibrated rank threshold, and adds only those classes which satisfy both conditions to the prediction set. $k$-CCP degenerates to CCP in the worst-case when the label rank threshold for each class is set to the total number of classes. Intuitively, the reduction is prediction set sizes from $k$-CCP over CCP depends on how small the label rank thresholds are which is classifier-dependent. We prove that $k$-CCP guarantees class-conditional coverage for each class and produces smaller prediction set sizes when compared to the CCP baseline. Our experiments on multiple benchmark datasets demonstrate that $k$-CCP produces smaller prediction sets over the baseline CCP method.

**Contributions.** The main contributions of this paper are as follows.

- Design of a novel $k$-CCP algorithm for class-conditional coverage by calibrating a pair of thresholds, one based on the conformity score, one based on the ranking of the score of the classifier, for each class to exploit the top-$k$ accuracy of the given classifier.
- Developing a theoretical analysis to demonstrate the failure of marginal CP, to prove that $k$-CCP guarantees class-conditional coverage and produces smaller prediction sets.
- Performing experimental evaluation of $k$-CCP on multiple imbalanced data benchmarks to demonstrate its efficacy over prior CP methods. Our anonymized code is in the Appendix.

## 2 PROBLEM SETUP AND NOTATIONS

We consider the problem of uncertainty quantification of classifiers for imbalanced data using the CP framework. Suppose $(X, Y)$ is a data sample where $X$ is an input from the input space $\mathcal{X}$ and $Y \in \mathcal{Y} = \{1, 2, \cdots, C\}$ is the ground-truth label, where $C$ is the number of candidate classes. Assume all data can be treated as random variables drawn from an underlying distribution $\mathcal{P}$ defined on $\mathcal{X} \times \mathcal{Y}$. We consider imbalanced data settings where the frequency distribution of class labels exhibits long-tail. Let $f$ denote a soft classifier (produces scores for all candidate classes, e.g., softmax scores) trained on a given training set of $m$ examples $\mathcal{D}_{\text{tr}} = \{(X_1, Y_1), \cdots, (X_m, Y_m)\}$, and $\mathcal{D}_{\text{tr}}^c = \{(X_i, Y_i) : Y_i = c\}$ denotes the data from class $c$ where $m_c = |\mathcal{D}_{\text{tr}}^c|$ is the number of training examples for class $c$. We are provided with a calibration set $\mathcal{D}_{\text{cal}} = \{(X_1, Y_1), \cdots, (X_n, Y_n)\}$ to enable CP. Let $\mathcal{I}_c = \{i : Y_i = c, \text{ for all } (X_i, Y_i) \in \mathcal{D}_{\text{cal}}\}$ and $n_c = |\mathcal{I}_c|$ is the number of calibration examples for class $c$. We define the top-$k$ error for class $c$ of the trained classifier $f$ as $\epsilon_c^k = \mathbb{P}\{r_f(X, Y) > k | Y = c\}$, where $r_f(x, y) = \sum_{c=1}^{C} \mathbb{1}[f(x)_c \geq f(x)_y]$ gives the rank of $y$ in prediction $f(x)$, where $\mathbb{1}[\cdot]$ is an indicator function.

**Problem Definition.** Our goal is to study provable CP methods that achieve class-conditional coverage and produce small prediction sets. Specifically, given a user-specified target coverage $1 - \alpha$, our aim is to construct small prediction sets $\widehat{\mathcal{C}}(X)$ to achieve class-conditional coverage for any

$c \in \mathcal{Y} = \{1, 2, \cdots, C\}$ on population level, where probability is over both calibration and test sets:

$$\mathbb{P}\{Y \in \widehat{\mathcal{C}}(X) | Y = c\} \geq 1 - \alpha \tag{1}$$

## 3 Failure of Marginal Conformal Prediction

**Marginal CP (MCP).** Let $V : \mathcal{X} \times \mathcal{Y} \to \mathbb{R}$ denote a *non-conformity* scoring function to measure how different a new example is from old examples (Vovk *et al.*, 2005). It is employed to compare a given testing input $(X, Y)$ with a set of calibration data $\mathcal{D}_{\text{cal}}$: if the non-conformity score is large, the new example $(X, Y)$ conforms less to calibration samples. For simplicity of notation, we denote the corresponding non-conformity score of the $i$-th calibration example as $V_i = V(X_i, Y_i)$.

For a target coverage $1 - \alpha$, we find the empirical quantile on calibration data $\mathcal{D}_{\text{cal}}$ defined as

$$\widehat{Q}_{1-\alpha}^{\text{MCP}} = \min\Big\{ t : \sum_{i=1}^{n} \frac{1}{n} \cdot \mathbb{1}[V_i \leq t] \geq 1 - \alpha \Big\} \tag{2}$$

$\widehat{Q}_{1-\alpha}^{\text{MCP}}$ can be determined by finding the $\lceil (1-\alpha)(1+n) \rceil$-th smallest value of $\{V_i\}_{i=1}^{n}$. The prediction set of a new testing input $X_{n+1}$ can be constructed by thresholding with $\widehat{Q}_{1-\alpha}^{\text{MCP}}$:

$$\widehat{\mathcal{C}}_{1-\alpha}^{\text{MCP}}(X_{n+1}) = \{y \in \mathcal{Y} : V(X_{n+1}, y) \leq \widehat{Q}_{1-\alpha}^{\text{MCP}}\} \tag{3}$$

Prior work has considered the design of good non-conformity scoring functions $V$, e.g., (Angelopoulos and Bates, 2021; Shafer and Vovk, 2008; Romano *et al.*, 2020) which produce small prediction sets. (Romano *et al.*, 2020) has proposed APS score for classification tasks based on ordered probabilities that is defined as follows. For a given input $X$, we sort the probabilities for all classes $\{1, \cdots, C\}$ using the classifier $f$ such that $1 \geq f(X)_{(1)} \geq \cdots \geq f(X)_{(C)} \geq 0$, where $f(x)_{(c)}$ denotes the $c$-th largest prediction. Then APS score for a sample $(x, y)$ can be computed as follows:

$$V(x, y) = f(x)_{(1)} + \cdots + f(x)_{(r_f(x,y)-1)} + U \cdot f(x)_{(r_f(x,y))} \tag{4}$$

where $U \in [0, 1]$ is a random variable drawn from uniform distribution to break ties and $r_f(x, y) = \sum_{c=1}^{C} \mathbb{1}[f(x)_c \geq f(x)_y]$ is the rank for $y$ of $f(x)$ in a descending order. Using the above definition of $V$ in (4), $\widehat{\mathcal{C}}^{\text{MCP}}$ (3) gives a marginal coverage guarantee (Romano *et al.*, 2020):

$$P\{Y \in \widehat{\mathcal{C}}_{1-\alpha}^{\text{MCP}}(X)\} \geq 1 - \alpha \tag{5}$$

Below we show that even under a ubiquitous condition, MCP results in over or under-coverage on some classes, where under-coverage violates the class-conditional coverage constraint.

**Failure Analysis of MCP.** To analyze the class-conditional coverage performance of MCP, we first define the class-wise empirical quantile for a given coverage probability $1 - \alpha$ as follows:

$$\widehat{Q}_{1-\alpha}^{\text{class}}(y) = \min\Big\{ t : \sum_{i \in \mathcal{I}_y} \frac{1}{n_y} \cdot \mathbb{1}[V(X_i, Y_i) \leq t] \geq 1 - \alpha \Big\} \tag{6}$$

Moreover, we need to define a set of robust classes for a fixed $\alpha$, i.e., $\text{Rob}(\alpha) := \{y \in \mathcal{Y} : \widehat{Q}_{1-\alpha}^{\text{class}}(y) \leq \widehat{Q}_{1-\alpha}^{\text{MCP}}\}$, which includes all classes whose class-wise empirical quantile $\widehat{Q}_{1-\alpha}^{\text{class}}(y)$ is smaller than the marginal empirical quantile $\widehat{Q}_{1-\alpha}^{\text{MCP}}$.

**Proposition 1.** *(Class-conditional over- and under-coverage of MCP) Given $\alpha$, assume $|Rob(\alpha)| < C$. If there exist $\xi, \xi' > 0$ such that for $y \in Rob(\alpha), y' \notin Rob(\alpha)$:*

$$\mathbb{P}\{V(X, Y) \leq \widehat{Q}_{1-\alpha}^{class}(Y) | Y = y \in Rob(\alpha)\} - \mathbb{P}\{V(X, Y) \leq \widehat{Q}_{1-\alpha}^{MCP} | Y = y \in Rob(\alpha)\} \leq -\xi,$$

$$\mathbb{P}\{V(X, Y) \leq \widehat{Q}_{1-\alpha}^{class}(Y) | Y = y' \notin Rob(\alpha)\} - \mathbb{P}\{V(X, Y) \leq \widehat{Q}_{1-\alpha}^{MCP} | Y = y' \notin Rob(\alpha)\}$$
$$\geq 1/n_{y'} + \xi'. \tag{7}$$

*Then class $y$ and $y'$ are over- and under-covered, respectively:*

$$\mathbb{P}\{V(X, Y) \leq \widehat{Q}_{1-\alpha}^{MCP} | Y = y\} \geq 1 - \alpha + \xi, \quad \mathbb{P}\{V(X, Y) \leq \widehat{Q}_{1-\alpha}^{MCP} | Y = y'\} \leq 1 - \alpha - \xi'$$

---

**Algorithm 1** $k$-Class-Conditional Conformal Prediction ($k$-CCP)

---

1: **Input**: error level $\alpha \in (0, 1)$.
2: Randomly split data into train $\mathcal{D}_{\text{tr}}$ and calibration $\mathcal{D}_{\text{cal}}$ and train the classifier $f$ on $\mathcal{D}_{\text{tr}}$
3: **for** $c \in \{1, \cdots, C\}$ **do**
4:     Compute $\{V_i\}_{i=1}^{n_c}$ for all $(X_i, Y_i) \in \mathcal{D}_{\text{cal}}$ such that $Y_i = c$
5:     Configure nominated error $\tilde{\alpha}_c$ and calibrate label rank $\widehat{k}(c)$ as shown in (10) and (11)
6:     $\widehat{Q}_{1-\tilde{\alpha}}^{\text{class}}(c) \leftarrow \lceil (1 - \tilde{\alpha}_c)(1 + n_c) \rceil$-th smallest value in $\{V_i\}_{i=1}^{n_c}$ as shown in (6)
7: **end for**
8: Construct $\widehat{\mathcal{C}}_{1-\alpha}^{k\text{-CCP}}(X_{n+1})$ for a new testing input $X_{n+1}$ using (9)

---

**Remark 1.** The above result shows that the universal threshold $\widehat{Q}_{1-\alpha}^{\text{MCP}}$ derived from MCP can achieve poor class-conditional coverage on some classes, as long as (7) holds. The interpretation of condition in (7) is that some class-wise quantiles $\widehat{Q}_{1-\alpha}^{\text{class}}(y)$ deviate from (either smaller or larger than) the marginal quantile $\widehat{Q}_{1-\alpha}^{\text{MCP}}$ with a class-conditional coverage margin $\xi$ or $\xi'$ on corresponding classes. In fact, this condition happens very frequently, particularly on imbalanced classification data, including training with standard loss (e.g., cross-entropy loss) or balanced loss (e.g., (Cao *et al.*, 2019)). We empirically demonstrate the deviation of class-wise quantiles from the marginal quantile, on real-world datasets in Figure 1 (first column) of Section 5.2. In summary, this theoretical/empirical analysis result shows the critical challenge for achieving class-conditional coverage.

## 4 TOP-$k$ CLASS-CONDITIONAL CONFORMAL PREDICTION

In this section, we first describe the details of the proposed top-$k$ Class-Conditional Conformal Prediction (denoted by $k$-CCP) algorithm followed by its theoretical analysis.

### 4.1 ALGORITHM DESIGN

Before we discuss our proposed $k$-CCP algorithm, we briefly describe the key idea behind the existing class-conditional CP algorithm(Angelopoulos and Bates, 2021; Sadinle *et al.*, 2019; Vovk, 2012) (denoted by CCP throughout this paper) to motivate our algorithm design. Specifically, CCP performs class-wise calibration on $\mathcal{D}_{\text{cal}}$ by finding class-wise empirical quantiles $\widehat{Q}_{1-\alpha}^{\text{class}}(y)$, as defined in (6), for each $y \in \mathcal{Y}$. For a given testing input $X_{n+1}$, CCP constructs the prediction set by

$$\widehat{\mathcal{C}}_{1-\alpha}^{\text{CCP}}(X_{n+1}) = \{y \in \mathcal{Y} : V(X_{n+1}, y) \leq \widehat{Q}_{1-\alpha}^{\text{class}}(y)\} \tag{8}$$

The above equation (8) shows that, for a given testing input $X_{n+1}$, CCP iterates every single $y \in \mathcal{Y}$ to compare the non-conformity score $V(X_{n+1}, y)$ to $\widehat{Q}_{1-\alpha}^{\text{class}}(y)$: includes the corresponding class label $y$ into the prediction set if the quantile can cover the score $V(X_{n+1}, y)$. We argue that, under some mild conditions, this principle behind CCP which scans all $y \in \mathcal{Y}$ can result in large prediction sets which is detrimental to human-ML collaborative systems.

Our proposed $k$-CCP algorithm (summarized in Algorithm 1) avoids scanning all class labels $y \in \mathcal{Y}$ by leveraging good properties of the given classifier $f$ in terms of its top-$k$ accuracy. Intuitively, if $f$ is sufficiently accurate, then the top-$k$ predicted class labels of $f(X_{n+1})$ for a given testing input $X_{n+1}$ will likely cover the true label $Y_{n+1}$. Specifically, $k$-CCP performs thresholding using class-wise quantiles only on a small subset of classes, i.e., top-$k$ classes, based on the soft scores of the trained classifier $f$. Consequently, we are able to avoid including irrelevant labels which are ranked after top-$k$ into the prediction set, thereby significantly reducing the prediction set size. So higher the top-$k$ accuracy of classifier $f$, smaller the prediction sets from $k$-CCP over the CCP method.

However, a critical challenge for this algorithmic design choice is how to determine the appropriate value of $k$. $k$-CCP addresses this challenge by using two inter-related threshold-based conditions to include a candidate class label $y$ in the prediction set of a given input $X_{n+1}$: one for the conformity score $V(X_{n+1}, y)$ and another for the rank of the label $y$, $r_f(X_{n+1}, y)$. These two thresholds collectively enable different trade-offs between class-conditional coverage and prediction set size. Our proposed $k$-CCP algorithm makes specific choices to estimate these two thresholds to achieve optimized trade-offs. $k$-CCP estimates class-wise quantiles $\widehat{Q}_{1-\tilde{\alpha}}^{\text{class}}(y)$ for $y \in \mathcal{Y}$ and calibrates the

value of $k$ for each class $y$, $\widehat{k}(y)$ on $\mathcal{D}_{\text{cal}}$ in a class-wise manner. $k$-CCP constructs the prediction set for a given testing input $X_{n+1}$ as follows:

$$\widehat{\mathcal{C}}_{1-\alpha}^{k\text{-CCP}}(X_{n+1}) = \{y \in \mathcal{Y} : V(X_{n+1}, y) \leq \underbrace{\widehat{Q}_{1-\tilde{\alpha}_y}^{\text{class}}(y)}_{\text{inflated quantile}}, \underbrace{r_f(X_{n+1}, y) \leq \widehat{k}(y)}_{\text{calibrated label rank } k}\} \qquad (9)$$

Comparing (9) to (8), there are two different design ideas. First, $\widehat{Q}_{1-\tilde{\alpha}_y}^{\text{class}}$ is the class-wise quantile with corresponding probability $1 - \tilde{\alpha}_y$ in (9), rather than the class-wise quantile with $1 - \alpha$ in (8). We require $1 - \tilde{\alpha}_y \geq 1 - \alpha$, as an inflation on nominated coverage, so that the trade-off of the calibrated rank $\widehat{k}(y)$ can be enabled. Second, $k$-CCP additionally employs calibrated class-wise $\widehat{k}(y)$ to reduce the candidate labels considered for inclusion in the prediction set (8). We investigate how to choose this class-specific inflation and calibrated label ranks in Section 4.2 based on Theorem 1 and Remark 2. Therefore, $k$-CCP actually introduces a trade-off between the calibration of non-conformity scores and calibration of label ranks, so that $k$-CCP can reduce the prediction set size. $k$-CCP degenerates to CCP when $\widehat{k}(y)$ is set to $C$ for every class $y$. Intuitively, the reduction of prediction set sizes by $k$-CCP over CCP depends on how small the values of $\widehat{k}(y) \in [1, C]$. The $\widehat{k}(y)$ values depend on the classifier. In the worst-case when $\widehat{k}(y) = C$, $k$-CCP will exhibit the same behavior as CCP.

## 4.2 THEORETICAL ANALYSIS

In this section, we provide theoretical analysis of the $k$-CCP algorithm in terms of class-conditional coverage guarantee and reduced size of the prediction sets compared to the CCP algorithm.

**Class-conditional Coverage Analysis.** Before presenting the main result, we first introduce class-wise top-$k$ error for a given class-wise calibrated label rank $\widehat{k}(y)$, i.e., $\epsilon_y := \mathbb{P}_Z\{r_f(X, Y) > \widehat{k}(y)|Y = y\}$, where $Z = (X, Y)$ is the joint random variable over input-output pairs and we drop the superscript $k$ in $\epsilon_c^k$, since $\widehat{k}(y)$ is a function of $y$. We highlight that $\widehat{k}(y) \in [C]$ can be any integer from 1 to $C$, and $\epsilon_y$ decreases as $k(y)$ increases. Specifically, $\epsilon_y$ reduces to top-1 error if $\widehat{k}(y) = 1$ (the minimum rank), whereas $\epsilon_y = 0$ if $\widehat{k}(y) = C$ (the maximum rank). The following main result indicates the feasible range of $\epsilon_y$, and thus the feasible range of $\widehat{k}(y)$.

**Theorem 1.** *($k$-CCP guarantees class-conditional coverage) Suppose that selecting $\{\widehat{k}(y)\}_{y \in \mathcal{Y}}$ results in class-wise top-k errors $\{\epsilon_y\}_{y \in \mathcal{Y}}$. For a target class-conditional coverage 1-$\alpha$, if the nominated mis-coverage probability $\tilde{\alpha}_y$ of $k$-CCP for class $y$ is set as follows*

$$\tilde{\alpha}_y \leq \alpha - \varepsilon_{n_y} - \delta - \epsilon_y, \quad \text{for } 0 < \delta < 1, \ \varepsilon_{n_y} = \sqrt{(3(1-\alpha)\log(2/\delta))/n_y}, \qquad (10)$$

*then $k$-CCP can achieve the class-conditional coverage as defined in equation (1), where $n_y$ is the number of calibration examples whose class label is $y$.*

**Remark 2.** The above result shows that the inflation of the nominated coverage from the target $1-\alpha$ in class-conditional coverage of (1) to the nominated level $1 - \tilde{\alpha}_y$ of $k$-CCP for class $y$ is bounded by $\varepsilon_{n_y} + \epsilon_y + \delta$. This implies that $k$-CCP can use the $\epsilon_y$ to set $\tilde{\alpha}$ once $\widehat{k}(y)$ is determined for each class $y$. On the other hand, since $\tilde{\alpha} > 0$, we can also derive the feasible range of $\epsilon_y \leq \alpha - \varepsilon_{n_y} - \delta$, which means that we have to select a sufficiently large $\widehat{k}(y)$ , so that $\epsilon_y$ is small enough to satisfy this constraint. The effectiveness of $k$-CCP to produce small prediction sets over CCP depends on how small $\widehat{k}(y)$ values are. If $\widehat{k}(y) = C$, then $k$-CCP reduces to CCP. The reduction in prediction set size is proportional to how small $\widehat{k}(y)$ values are. Empirically, for each class $y$, we determine:

$$\widehat{k}(y) = \min\{c \in [C] : \frac{1}{n_y}\sum_{i \in \mathcal{I}_y} \mathbb{1}[r_f(X_i, Y_i) > c] \leq \alpha - g/\sqrt{n_y}\}, \qquad (11)$$

where $g \in \{0.1, 0.15, \cdots, 1\}$ is a hyper-parameter which is tuned on validation data in terms of small prediction sets. Indeed, our results in Figure 1 (the last column) demonstrate that classifiers trained on real-world datasets have small $\widehat{k}(y)$ values (shown in terms of $\sigma_y \in [0, 1]$ which is introduced below).

**Prediction Set Size Analysis.** After deriving the guarantee for the class-conditional coverage of $k$-CCP, we study under what conditions $k$-CCP can produce smaller expected prediction set size compared to the CCP baseline. For fair comparison, we assume both $k$-CCP and CCP algorithms guarantee $1-\alpha$ class-conditional coverage. For the result of class-conditional coverage for CCP, we refer readers to (Vovk, 2012; Sadinle *et al.*, 2019; Angelopoulos and Bates, 2021). Before presenting the main result, we define a condition number as follows:

$$\sigma_y = \mathbb{P}_{X_{n+1}}\Big[V(X_{n+1}, y) \leq \widehat{Q}^{\text{class}}_{1-\widetilde{\alpha}}(y),\ r_f(X_{n+1}, y) \leq \widehat{k}(y)\Big]\Big/\mathbb{P}_{X_{n+1}}\Big[V(X_{n+1}, y) \leq \widehat{Q}^{\text{class}}_{1-\alpha}(y)\Big]$$

Specifically, $\sigma_y$ represents the benefit for the trade-off between the coverage with inflated quantile $\widehat{Q}^{\text{class}}_{1-\widetilde{\alpha}}$ and the constraint with calibrated rank $\widehat{k}(y)$.

**Theorem 2.** *($k$-CCP produces smaller prediction sets than CCP) Suppose the following inequality holds for any $y \in \mathcal{Y}$:*

$$\sum_{y \in \mathcal{Y}} \sigma_y \cdot \mathbb{P}_{X_{n+1}}\Big[V(X_{n+1}, y) \leq \widehat{Q}^{class}_{1-\alpha}(y)\Big] \leq \sum_{y \in \mathcal{Y}} \mathbb{P}_{X_{n+1}}\Big[V(X_{n+1}, y) \leq \widehat{Q}^{class}_{1-\alpha}(y)\Big] \qquad (12)$$

*Then $k$-CCP produces smaller expected prediction sets than CCP, i.e.,*

$$\mathbb{E}_{X_{n+1}}[|\widehat{\mathcal{C}}^{k\text{-}CCP}_{1-\widetilde{\alpha}}(X_{n+1})|] \leq \mathbb{E}_{X_{n+1}}[|\widehat{\mathcal{C}}^{CCP}_{1-\alpha}(X_{n+1})|]$$

**Remark 3.** The above result demonstrates that for a target class-conditional coverage $1-\alpha$, when both $k$-CCP and CCP achieves the target class-conditional coverage, under the condition of (12), $k$-CCP produces smaller prediction sets than CCP. Now we can interpret the term $\sigma_y$ as class-specific weights for aggregating the coverage using $\widehat{Q}^{\text{class}}_{1-\alpha}$. In (12), when the $\sigma_y$-weighted coverage aggregation (left side) is smaller than the uniform one (right side), $k$-CCP produces smaller prediction sets when compared to CCP. Our comprehensive experiments (Section 5.2, Figure 1, the last column) show $\sigma_y$ values are much smaller than 1 for all datsets, so that demonstrate the practical utility of our theoretical analysis to produce small prediction sets using $k$-CCP. Note that the reduction in prediction set size of $k$-CCP over CCP is proportional to how small the $\sigma_y$ values are, which are classifier-dependent.

# 5 EXPERIMENTS AND RESULTS

## 5.1 EXPERIMENTAL SETUP

**Classification datasets.** We consider four datasets for evaluation: CIFAR-10, CIFAR-100 (Krizhevsky *et al.*, 2009), mini-ImageNet (Vinyals *et al.*, 2016), and Food-101 (Bossard *et al.*, 2014) using the standard training and validation split. We employ the same methodology as (Menon *et al.*, 2020; Cao *et al.*, 2019; Cui *et al.*, 2019) to create an imbalanced long-tail setting for each dataset: 1) We use the original training split as training set for training $f$ with $N_{\text{tr}}$ samples, and randomly split the original (balanced) validation set into $N_{\text{cal}}$ calibration samples and $N_{\text{test}}$ testing samples. 2) We define an imbalance ratio $\rho$, the ratio between the sample size of the smallest and largest class: $\rho = \frac{\min_i \{\# \text{ samples in class } i\}}{\max_i \{\# \text{ samples in class } i\}}$. 3) For each training set, we create three different imbalanced distributions using three decay types over the class indices $c \in \{1, \cdots, C\}$: (a) An exponential-based decay (EXP) with $\frac{N_{tr}}{C} \times \rho^{\frac{c}{C}}$ examples in class $c$, (b) A polynomial-based decay (POLY) with $\frac{N_{tr}}{C} \times \frac{1}{\sqrt{\frac{c}{10\rho}+1}}$ examples in class $c$, and (c) A majority-based decay (MAJ) with $\frac{N_{tr}}{C} \times \rho$ examples in classes $c > 1$. We keep the calibration and test set balanced and unchanged. We provide an illustrative examples of the three decay types in Appendix (Section C.3, Figure 3).

**Deep Neural Network Models.** We consider ResNet-20 (He *et al.*, 2016) as the main architecture to train classifiers. To handle imbalanced data, we employ the training algorithm "LDAM" proposed by (Cao *et al.*, 2019) that assigns different margins to classes, where larger margins are assigned to minority classes in the loss function. We also add the conformalized training (Stutz *et al.*, 2021) [1] for CIFAR-100 dataset to demonstrate synergistic benefits of $k$-CCP. These results are summarized in Table 17 of Appendix F. We will add more results in final paper.

**CP Baselines.** We consider four CP methods: **1)** MCP that targets marginal coverage as in Euqation (3); **2)** CCP which estimates class-wise score thresholds and produces prediction set using Equation (8); **3)** cluster-CP (Ding *et al.*, 2023) [2] that performs calibration over clusters to reduce predic-

---

[1] https://github.com/google-deepmind/conformal_training/tree/main

[2] https://github.com/tiffanyding/class-conditional-conformal/tree/main

tion set sizes; and **4)** $k$-CCP that produces prediction set using Equation (9). All CP methods are built on the same classifier and non-conformity scoring function (either APS (Romano *et al.*, 2020) or RAPS (Angelopoulos *et al.*, 2020) for a fair comparison. We tune the hyper-parameters for each baselines according to their recommended ranges based on the same criterion (See Appendix C.2 for details). We repeat experiments over 10 different random calibration-testing splits and report the average performance with standard deviation.

**Evaluation methodology.** We use the target coverage $1 - \alpha = 90\%$ (marginal coverage for MCP and class-conditional coverage for CCP and $k$-CCP). We compute two metrics on the testing set:

• *Under Coverage Ratio (UCR).* $\text{UCR} := \sum_{c \in [C]} \mathbb{1}\left[\frac{\sum_{(x,y) \in \mathcal{D}_{\text{test}}} \mathbb{1}[y \in \widehat{\mathcal{C}}(x) \text{ s.t. } y=c]}{\sum_{(x,y) \in \mathcal{D}_{\text{test}}} \mathbb{1}[y=c]} < 1 - \alpha\right]/C.$

• *Average Prediction Set Size (APSS).* $\text{APSS} = \sum_{c \in [C]} \frac{\sum_{(x,y) \in \mathcal{D}_{\text{test}}} \mathbb{1}[y=c] \cdot |\widehat{\mathcal{C}}(x)|}{\sum_{(x,y) \in \mathcal{D}_{\text{test}}} \mathbb{1}[y=c]}/C.$

For the three class-conditional CP algorithms, i.e., CCP, cluster-CP, and our $k$-CCP, we control their UCR as the same value that is close to 0 for a fair comparison of APSS. To this end, we uniformly add $g/\sqrt{n}$ to inflate the nominated coverage $1 - \alpha$ to each baseline, and tune $g \in \{0.1, 0.15, \cdots, 1\}$ on validation in terms of prediction set size. The actual achieved UCR values are shown in the complete results (see Table 15 and 16 of Appendix D and E).

## 5.2 RESULTS AND DISCUSSION

We list empirical results in Table 1 for an overall comparison on all four datasets with $\rho = 0.5, 0.1$ using all three training distributions (EXP, POLY and MAJ) based on the considered APS and RAPS

| Scoring Function | Methods | EXP | | POLY | | MAJ | |
|---|---|---|---|---|---|---|---|
| | | $\rho = 0.5$ | $\rho = 0.1$ | $\rho = 0.5$ | $\rho = 0.1$ | $\rho = 0.5$ | $\rho = 0.1$ |
| | | | | CIFAR-10 | | | |
| APS | MCP | $1.132 \pm 0.033$ | $1.406 \pm 0.045$ | $1.117 \pm 0.028$ | $1.214 \pm 0.038$ | $1.196 \pm 0.032$ | $2.039 \pm 0.046$ |
| | CCP | $1.481 \pm 0.082$ | $\mathbf{2.032 \pm 0.096}$ | $\mathbf{1.487 \pm 0.090}$ | $\mathbf{1.945 \pm 0.087}$ | $\mathbf{1.765 \pm 0.093}$ | $\mathbf{2.964 \pm 0.123}$ |
| | cluster-CP | $\mathbf{1.445 \pm 0.017}$ | $2.323 \pm 0.015$ | $1.612 \pm 0.013$ | $2.102 \pm 0.015$ | $1.787 \pm 0.019$ | $2.969 \pm 0.025$ |
| | $k$-**CCP** | $1.481 \pm 0.082$ | $\mathbf{2.032 \pm 0.096}$ | $\mathbf{1.487 \pm 0.090}$ | $\mathbf{1.945 \pm 0.087}$ | $\mathbf{1.765 \pm 0.093}$ | $\mathbf{2.964 \pm 0.123}$ |
| RAPS | MCP | $1.143 \pm 0.004$ | $1.419 \pm 0.013$ | $1.118 \pm 0.004$ | $1.233 \pm 0.006$ | $1.196 \pm 0.032$ | $2.043 \pm 0.016$ |
| | CCP | $1.502 \pm 0.007$ | $\mathbf{2.049 \pm 0.013}$ | $1.558 \pm 0.010$ | $\mathbf{1.776 \pm 0.012}$ | $\mathbf{1.786 \pm 0.020}$ | $\mathbf{2.628 \pm 0.012}$ |
| | cluster-CP | $\mathbf{1.493 \pm 0.017}$ | $2.323 \pm 0.015$ | $1.612 \pm 0.013$ | $1.981 \pm 0.013$ | $1.787 \pm 0.019$ | $2.968 \pm 0.024$ |
| | $k$-**CCP** | $1.502 \pm 0.007$ | $\mathbf{2.049 \pm 0.013}$ | $\mathbf{1.558 \pm 0.010}$ | $\mathbf{1.776 \pm 0.012}$ | $\mathbf{1.786 \pm 0.020}$ | $\mathbf{2.628 \pm 0.012}$ |
| | | | | CIFAR-100 | | | |
| APS | MCP | $10.303 \pm 0.111$ | $14.544 \pm 0.119$ | $15.729 \pm 0.126$ | $25.888 \pm 0.197$ | $11.680 \pm 0.117$ | $23.796 \pm 0.159$ |
| | CCP | $44.194 \pm 0.514$ | $50.963 \pm 0.481$ | $49.895 \pm 0.489$ | $64.366 \pm 0.389$ | $48.323 \pm 0.548$ | $64.640 \pm 0.621$ |
| | cluster-CP | $30.922 \pm 0.454$ | $43.883 \pm 1.070$ | $56.696 \pm 0.393$ | $63.208 \pm 0.364$ | $33.623 \pm 0.395$ | $50.382 \pm 0.711$ |
| | $k$-**CCP** | $\mathbf{20.355 \pm 0.357}$ | $\mathbf{25.185 \pm 0.278}$ | $\mathbf{25.843 \pm 0.300}$ | $\mathbf{37.034 \pm 0.244}$ | $\mathbf{21.196 \pm 0.320}$ | $\mathbf{35.630 \pm 0.232}$ |
| RAPS | MCP | $10.300 \pm 0.080$ | $14.554 \pm 0.107$ | $15.755 \pm 0.103$ | $25.850 \pm 0.150$ | $11.684 \pm 0.091$ | $23.708 \pm 0.137$ |
| | CCP | $44.243 \pm 0.340$ | $50.969 \pm 0.345$ | $49.877 \pm 0.354$ | $64.247 \pm 0.234$ | $48.337 \pm 0.355$ | $64.580 \pm 0.536$ |
| | cluster-CP | $30.971 \pm 0.454$ | $43.883 \pm 1.073$ | $56.656 \pm 0.354$ | $63.113 \pm 0.397$ | $33.656 \pm 0.388$ | $50.365 \pm 0.701$ |
| | $k$-**CCP** | $\mathbf{20.355 \pm 0.005}$ | $\mathbf{25.185 \pm 0.011}$ | $\mathbf{25.843 \pm 0.006}$ | $\mathbf{37.035 \pm 0.005}$ | $\mathbf{21.197 \pm 0.005}$ | $\mathbf{35.631 \pm 0.007}$ |
| | | | | mini-ImageNet | | | |
| APS | MCP | $9.705 \pm 0.101$ | $8.930 \pm 0.093$ | $9.810 \pm 0.101$ | $9.665 \pm 0.101$ | $9.840 \pm 0.091$ | $9.123 \pm 0.086$ |
| | CCP | $26.666 \pm 0.415$ | $34.867 \pm 0.445$ | $26.620 \pm 0.369$ | $29.852 \pm 0.360$ | $27.306 \pm 0.377$ | $29.200 \pm 0.379$ |
| | cluster-CP | $27.786 \pm 0.307$ | $33.114 \pm 0.418$ | $21.273 \pm 0.369$ | $25.550 \pm 0.279$ | $25.288 \pm 0.226$ | $25.229 \pm 0.352$ |
| | $k$-**CCP** | $\mathbf{18.129 \pm 0.453}$ | $\mathbf{17.769 \pm 0.463}$ | $\mathbf{17.784 \pm 0.438}$ | $\mathbf{19.153 \pm 0.412}$ | $\mathbf{18.110 \pm 0.414}$ | $\mathbf{18.594 \pm 0.439}$ |
| RAPS | MCP | $9.703 \pm 0.076$ | $9.003 \pm 0.067$ | $9.806 \pm 0.079$ | $9.714 \pm 0.075$ | $9.865 \pm 0.060$ | $9.146 \pm 0.063$ |
| | CCP | $26.689 \pm 0.177$ | $29.750 \pm 0.219$ | $21.352 \pm 0.196$ | $26.266 \pm 0.218$ | $36.535 \pm 0.196$ | $25.641 \pm 0.217$ |
| | cluster-CP | $27.466 \pm 0.268$ | $32.991 \pm 0.434$ | $21.212 \pm 0.298$ | $36.061 \pm 0.475$ | $32.085 \pm 0.424$ | $25.269 \pm 0.375$ |
| | $k$-**CCP** | $\mathbf{15.101 \pm 0.003}$ | $\mathbf{18.418 \pm 0.003}$ | $\mathbf{15.331 \pm 0.003}$ | $\mathbf{17.465 \pm 0.003}$ | $\mathbf{17.388 \pm 0.003}$ | $\mathbf{17.167 \pm 0.004}$ |
| | | | | Food-101 | | | |
| APS | MCP | $9.570 \pm 0.076$ | $13.998 \pm 0.089$ | $12.267 \pm 0.079$ | $16.468 \pm 0.095$ | $9.964 \pm 0.078$ | $23.796 \pm 0.159$ |
| | CCP | $40.408 \pm 0.378$ | $60.762 \pm 0.531$ | $45.148 \pm 0.342$ | $65.6723 \pm 0.515$ | $41.453 \pm 0.335$ | $66.633 \pm 0.622$ |
| | cluster-CP | $28.828 \pm 0.294$ | $44.885 \pm 0.589$ | $32.873 \pm 0.307$ | $38.326 \pm 0.248$ | $33.258 \pm 0.450$ | $46.430 \pm 0.337$ |
| | $k$-**CCP** | $\mathbf{17.281 \pm 0.225}$ | $\mathbf{20.610 \pm 0.222}$ | $\mathbf{20.452 \pm 0.209}$ | $\mathbf{24.771 \pm 0.192}$ | $\mathbf{19.398 \pm 0.223}$ | $\mathbf{26.584 \pm 0.191}$ |
| RAPS | MCP | $9.580 \pm 0.037$ | $14.039 \pm 0.055$ | $12.327 \pm 0.046$ | $16.541 \pm 0.060$ | $10.040 \pm 0.051$ | $16.293 \pm 0.047$ |
| | CCP | $40.411 \pm 0.285$ | $60.790 \pm 0.395$ | $36.550 \pm 0.141$ | $41.755 \pm 0.153$ | $32.957 \pm 0.224$ | $36.797 \pm 0.139$ |
| | cluster-CP | $28.919 \pm 0.287$ | $44.583 \pm 0.667$ | $32.928 \pm 0.358$ | $41.785 \pm 0.220$ | $32.983 \pm 0.518$ | $46.078 \pm 0.312$ |
| | $k$-**CCP** | $\mathbf{17.282 \pm 0.004}$ | $\mathbf{20.610 \pm 0.006}$ | $\mathbf{20.452 \pm 0.002}$ | $\mathbf{24.771 \pm 0.004}$ | $\mathbf{19.398 \pm 0.006}$ | $\mathbf{25.163 \pm 0.002}$ |

Table 1: APSS results comparing MCP, CCP, cluster-CP, and $k$-CCP with ResNet-20 model under different imbalance ratio $\rho = 0.5$ and $\rho = 0.1$. For a fair comparison of prediction set size, we set UCR of $k$-CCP the same as or smaller (more restrictive) than that of CCP and cluster-CP under 0.16 on CIFAR-10 and 0.03 on other datasets. The specified UCR values are in Table 15 and 16 of Appendix D and E. The APSS results show that $k$-CCP significantly outperforms CCP and cluster-CP in terms of average prediction set size.

score functions. Complete experiment results under more values of $\rho$ are in Appendix C). We make the following observations: (i) MCP does not provide class-conditional coverage, which is straightforward; (ii) CCP, cluster-CP, and $k$-CCP can guarantee the class-conditional coverage (their UCRs are all close to 0); and (iii) $k$-CCP significantly outperforms CCP and cluster-CP in APSS on almost all settings ($k$-CCP still outperforms others in most cases even on CIFAR-10).

To investigate the challenge of imbalanced data and more importantly how $k$-CPP significantly improves the APSS, we further conduct four justification experiments. We report the results on mini-ImageNet and Food-101 below and the complete ones in Appendix C. First, we visualize the distribution of class-wise quantiles compared to the marginal quantile (derived by MCP), highlighting the large variance of class-wise quantiles on imbalanced data. Second, we report the histograms of class-conditional coverage and the corresponding histograms of prediction set size. This experiment verifies that $k$-CCP derives significantly more class-conditional coverage above $1 - \alpha$ and thus reduces the prediction set size. Third, we empirically verify the trade-off condition number $\{\sigma_y\}_{y=1}^{C}$ in Theorem 2 to reveal the underlying reason for $k$-CCP producing smaller prediction sets over CCP. Finally, We add the ablation study to verify how hyper-parameter $g$ affects the performance of CCP, cluster-CP, and $k$-CCP. We set the range of $g$ from $\{0.1, 0.15, \cdots, 0.7\}$ on four datasets with imbalance ratio $\rho = 0.1$ EXP. Below we discuss our results and findings in more detail.

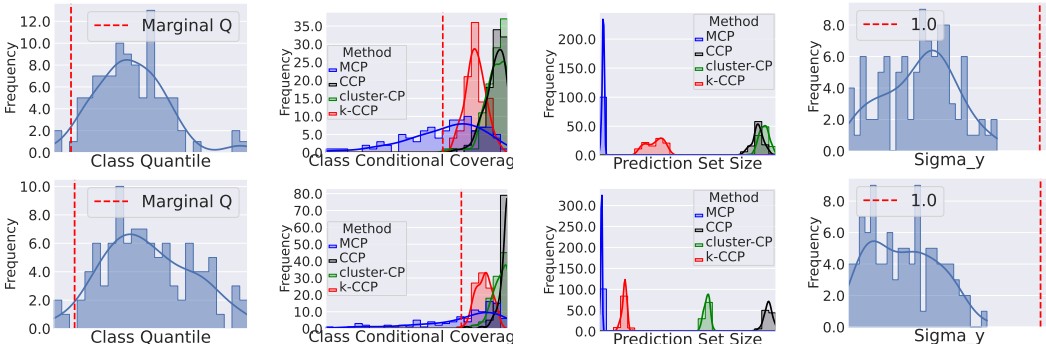

Figure 1: Justification experiments: mini-ImageNet in first row and Food-101 in second row with ResNet20 model. First column: distribution of class-wise quantiles v.s. marginal quantile with imbalance type EXP and imbalance ratio $\rho = 0.5$. Second and third columns: histograms of class-conditional coverage and prediction set size achieved by MCP, CCP, cluster-CP, and $k$-CCP with imbalance type EXP and imbalance ratio $\rho = 0.1$. The final column: the histogram of condition numbers $\sigma_y$ in Theorem 2 with imbalance type EXP and imbalance ratio $\rho = 0.1$.

**MCP does not guarantee class-conditional coverage.** From Table 1, MCP achieves a large UCR, indicating its failure as described in Proposition 1. Recall that Proposition 1 indicates that MCP has over or under-coverage on the classes where the corresponding class-wise quantiles deviate from the marginal one. To further verify this condition, we compare the distribution of class-conditional quantiles with the marginal quantile in histograms in the first column of Figure 1, which verifies that this condition easily holds, even for relatively balanced datasets (i.e., $\rho = 0.5$).

$k$**-CCP significantly outperforms CCP and Cluster-CP.** First, it is clear from Table 1 that $k$-CCP, CCP, and cluster-CP guarantee class-conditional coverage on all settings. This can also be shown by the second column of Figure 1, where the class-conditional coverage bars of CCP and $k$-CCP distribute on the right-hand side of the target probability $1 - \alpha$ (red dashed line). Second, $k$-CCP outperforms CCP with a large margin in terms of average prediction set size under the same level of class-conditional coverage. We also report the histograms of the corresponding prediction set sizes in the third column of Figure 1, which shows (i) $k$-CCP has more concentrated class-conditional coverage distribution than CCP and cluster-CP; (ii) the distribution of prediction set sizes produced by $k$-CCP is globally smaller than that produced by CCP and cluster-CP. This observation can be justified by a better trade-off number of $\{\sigma_y\}_{y=1}^{C}$ as shown below.

**Verification of $\sigma_y$.** The last column of Figure 1 verifies the validity of Theorem 2 on testing dataset and confirms the optimized trade-off between the coverage with inflated quantile and the constraint with calibrated rank leads to smaller prediction sets. Experiments even show a stronger condition ($\sigma_y$ is much less than 1 for all $y$) than the weighted aggregation condition in (12). We notice that

$k$-CCP reduces to CCP on CIFAR-10, so $\sigma_y = 1$ for all $y$ and there is no trade-off. On other three datasets, we satisfy the conditions needed by $k$-CCP. In addition, we also verify the assumption of $\sigma_y$ firmly holds on calibration sets in Figure 24 of Appendix I.

**Sensitivity for $g$.** Figure 2 shows the sensitivity of CCP, cluster-CP, and $k$-CCP for $g$ on mini-ImageNet and Food-101 with APS scoring function. It is clear that the UCR and APSS of $k$-CCP are much smaller than CCP and cluster-CP with the same $g$ values on CIFAR-100, mini-ImageNet, and Food-101. These results verify that $k$-CCP achieves a better trade-off between coverage and prediction set size. In addition, we also study the the sensitivity of CCP, cluster-CP, and $k$-CCP with RAPS scoring function in Figure 22 of Appendix H.

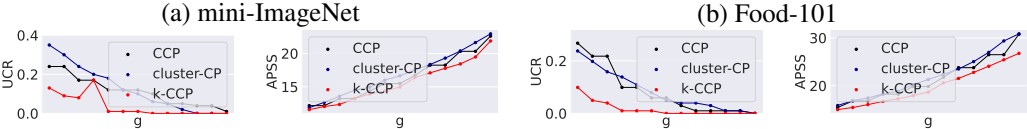

Figure 2: Results for under coverage ratio and average prediction set size achieved by CCP, cluster-CP, and $k$-CCP methods as a function of $g$ using APS scoring function with imbalance type EXP for imbalance ratio $\rho = 0.1$.

## 6 RELATED WORK

**Learning Classifiers for Imbalanced Data.** Classification algorithms for imbalanced datasets can be categorized into two main categories: 1) *Re-sampling*. This category aims to balance the population of the majority class and the minority class by over-sampling (Chawla *et al.*, 2002; Mohammed *et al.*, 2020; Krawczyk *et al.*, 2019) or under-sampling (Tsai *et al.*, 2019; Vuttipittayamongkol and Elyan, 2020). 2) *Re-weighting*. This category aims to overcome the imbalanced data challenges by assigning adaptive weights to different classes/samples (Huang *et al.*, 2019; Madabushi *et al.*, 2020). There is also work on providing theoretical analysis for the trained classifiers (Cao *et al.*, 2019; Gottlieb *et al.*, 2021). However, prior work on learning classifiers does not consider the problem of long-tailed label-distributions (i.e., imbalanced data) from an uncertainty quantification perspective.

**Conformal Prediction.** CP (Vovk *et al.*, 1999; 2005; Romano *et al.*, 2020) is a general framework for uncertainty quantification that provides marginal coverage guarantees without any assumptions on the underlying data distribution. Prior work has considered improving on the standard CP to handle distribution shifts that may be caused by long-term distribution shift (Gibbs and Candes, 2021), covariate shift (Tibshirani *et al.*, 2019), or label-distribution shift (Podkopaev and Ramdas, 2021). However, there is little work on studying instantiations of the CP framework to adapt to the imbalanced data setting (Vovk, 2012; Sadinle *et al.*, 2019; Angelopoulos and Bates, 2021). The only known CP framework that guarantees class-conditional coverage was proposed in (Vovk, 2012; Sadinle *et al.*, 2019) and surveyed by (Angelopoulos and Bates, 2021). It builds one conformal predictor for each class resulting in large prediction sets. Clustered CP (Ding *et al.*, 2023) groups class labels into clusters and performs calibration over clusters to reduce prediction set sizes. However, it does not provide theoretical guarantees on class-conditional coverage. Note that conditional CP methods for *input space* (Gibbs *et al.*, 2023; Feldman *et al.*, 2021) are not applicable to *output space*, i.e., the class-conditional setting. Our provable $k$-CCP approach leverages the ranking of candidate class labels by the classifier to configure the calibration process to produce small prediction sets.

## 7 SUMMARY

This paper studies a provable conformal prediction (CP) algorithm that aims to provide class-conditional coverage guarantee and produce small prediction sets for the challenging imbalanced data settings. Our proposed $k$-CCP algorithm performs double-calibration, one over conformity score and one over label rank for each class separately, to achieve this goal by leveraging the beneficial properties of top-$k$ accuracy of a trained classifier. Our extensive experiments clearly demonstrate the significant efficacy of $k$-CCP over the baseline CCP algorithm.

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
