# OpenReview forum: "Class-Conditional Conformal Prediction for Imbalanced Data via Top-$k$ Classes"
_ICLR.cc/2024/Conference — Submitted to ICLR 2024_

### Official Review · Reviewer_zE7U · 2023-10-22

**Soundness:** 3 good
**Presentation:** 2 fair
**Contribution:** 2 fair
**Rating:** 3
**Confidence:** 2

**Summary:**

This paper is motivated by multiclass imbalanced classification, where some classes are very rare. Often, classifiers optimizing for accuracy or similar metrics do not predict such classes ever. This paper proposes k-Class-Conditional Conformal Prediction (k-CCP). The main idea is to modify Class-Conformal Prediction by only considering the top k classes from the classifier. The main goal is to ensure that the prediction set is not too large. Theoretical and numerical results are provided.

**Strengths:**

* The paper provides three discrete contributions: (1) a theorem that standard conformal prediction (here, marginal conformal prediction), is insufficient for class conformal prediction (CCP), (2) the k-CCP algorithm, and (3) a theoretical analysis of k-CCP.
* The paper is fairly clear.

**Weaknesses:**

* The value of the k-CCP contribution over CCP is unclear. It seems relatively modest to restrict to the top-k classes. Additionally, the importance of Theorem 2 and the numerical results are unclear to me. For the theorem, the condition is fairly unintuitive. For the numerical results, e.g., Table 1, I'm not sure that setting the UCR of k-CCP and CCP to be the same necessarily leads to a fair comparison. At some point, it seems like you have to trade off class conditioned coverage and prediction set size.

**Questions:**

* A concrete real-world example would be better. Most of the experiments are on vision, but the only application mentioned in the abstract and introduction are general references to medical diagnoses.
* A concrete example instead of or combined with Proposition 1 would help. Intuitively, it seems like conformal prediction with imbalanced classes would suffer the same problems as accuracy optimization--the classes that have few samples in training/calibration data are unlikely to be predicted. It seems like a more intuitive example could be provided.
* The algorithms could be written out more clearly without referencing theorems, remarks, and equations in the main text.
* There are some typos ("pupulation" p. 3, "K" for the number of classes p. 6).

---

> ### Author Response · Authors · 2023-11-19
> **Response For Reviewer zE7U**
>
> We thank the reviewer for their thoughtful comments and feedback.
>
> Q1: **The value of the k-CCP contribution over CCP is unclear. It seems relatively modest to restrict to the top-k classes.**
>
> A1: The $k$-CCP algorithm adapts the double calibration mechanism (calibration on both non-conformity score and ranks in Equation (9), instead of a single calibration in standard CCP) to take a good trade-off of coverage and efficiency. The key advantage of the additional calibration over rank is to exclude those data where their true labels are ranked beyond top-$\widehat k(y)$ positions (i.e., poorly predicted data) in the calibration process, which in turn helps remove labels beyond top-$\widehat k(y)$ positions in the testing process.
>
> Q2: **The importance of Theorem 2 and the numerical results are unclear.**
>
> A2: The contribution of Theorem 2 is to theoretically identify the source of reduced prediction set sizes by $k$-CCP, so that it enables verification in experimental results shown in Figure 1 (last column). We additionally verify it on calibration data in Figure 24 of the updated paper. We believe that Theorem 2 reveals this insight/knowledge and wraps up the key factors into the condition numbers $\sigma_y$.
>
> Our experiments on multiple benchmark datasets demonstrate the following three remarks.
>
> (i) MCP fails to provide class-conditional coverage in the first columns of Figure 1 and 4 .
>
> (2) $k$-CCP produces smaller prediction sets over the baseline CCP method in Table 1. We also added Table 15, 16, and 17 in our revised paper to compare $k$-CCP with cluster-CP [R20] (a newly added baseline for achieving approximate class-conditional coverage) with APS score function (Table 15) and RAPS score function [R21] (Table 16). Moreover, we perform comparison between $k$-CCP and the above CP algorithms using a conformalized training algorithm [R22] (Table 17).
>
> (3) The condition numbers $\sigma_y$ in Theorem 2 are valid and verified in the last columns of Figure 1 and 4. We also added Figure 24 in our revised paper that verifies these condition numbers on calibration data.
>
> Q3: **For the theorem, the condition is fairly unintuitive.**
>
> A3: The condition (12) in Theorem 1 in the revised paper  follows the algorithmic design in Algorithm 1, i.e., iterating over all classes, which results in the summation over $y \in \mathcal Y$ in (12). Within each element (a fixed $y$), we define a condition $\sigma_y$ as the source of the reduced prediction set size for $k$-CCP. This insight can be easily verified in experiments (The last columns of Figure 1 and 4) and theoretically supports smaller prediction sets from $k$-CCP.
>
> Q4: **For the numerical results, e.g., Table 1 (Table 15 and 16 in our revised paper), I'm not sure that setting the UCR of k-CCP and CCP to be the same necessarily leads to a fair comparison.**
>
> A4: It is necessary to set UCR to the same value (also close to $0$) for all methods, because it is a measure of the violation of class-conditional coverage. Our paper studies the efficiency of CP methods that guarantees the class-conditional coverage. If UCR is not close to $0$ (like MCP), we cannot verify the class-conditional coverage is achieved, so that it is consequently not meaningful to compare prediction set sizes as the measure of efficiency.
>
>
> Q5: **A concrete example instead of or combined with Proposition 1 would help. It seems like a more intuitive example could be provided.**
>
> A5: We would like Proposition 1 to be a general and easy-to-verify theoretical result. This is why we did not connect it to the learning observation (recall that a key advantage of CP is model-agnostic, i.e., invariant to the learning algorithm). Instead, we can easily interpret from Proposition 1 and experiments to verify the conditions. The first columns of Figure 1 and 4 show the ease of under-coverage of MCP over each class, which in fact provides a concrete illustration for Proposition 1.
>
> Q6: **The algorithms could be written out more clearly without referencing theorems, remarks, and equations in the main text.**
>
> A6: Thank you for your suggestion. We have revised $k$-CCP in the Algorithm 1 (in our updated paper) by linking Line 5 to the new Equation (11) in the updated paper, i.e., $\widehat k(y) = \min\{ c \in [C]: \frac{1}{n_y} \sum_{i \in \mathcal I_y}  1[ r_f(X_i, Y_i) > c ] \leq \alpha - g / \sqrt{n_y} \}$, where $1[]$ is the indicator function.
>
> [R20] Ding, T., Angelopoulos, A. N., Bates, S., Jordan, M. I., & Tibshirani, R. J. (2023). Class-Conditional Conformal Prediction With Many Classes. arXiv preprint arXiv:2306.09335.
>
> [R21] Angelopoulos, A., Bates, S., Malik, J., & Jordan, M. I. (2020). Uncertainty sets for image classifiers using conformal prediction. arXiv preprint arXiv:2009.14193.
>
> [R22] Stutz, D., Cemgil, A. T., & Doucet, A. (2021). Learning optimal conformal classifiers. arXiv preprint arXiv:2110.09192.

---

> > ### Comment · Reviewer_zE7U · 2023-11-22
> > **Read response**
> >
> > I thank the authors for the response, and I still remain unconvinced of the contribution.

---

### Official Review · Reviewer_KjCS · 2023-10-26

**Soundness:** 3 good
**Presentation:** 3 good
**Contribution:** 3 good
**Rating:** 6
**Confidence:** 4

**Summary:**

This paper proposes a method for achieving class conditional coverage for conformal prediction. This method uses the top-k error to form thresholds for each class for Conformal Prediction. They empirically demonstrate that the length of the sets generated by existing class conditional CP methods is suboptimal compared to the proposed methods' prediction sets.

I have updated my score to a 6 to reflect the answers in the rebuttal. With adding more baselines, I feel this paper would be an 8. The theoretical contributions are currently strong, and the empirical contributions are fair.

**Strengths:**

1. The investigation into the over-coverage and under coverage is a good addition to demonstrate the need for class conditional coverage methods to be generated.
2. The intuition of using the top-k classes to remove spurious classes is quite a nice and intuitive idea. Moreover, besides the choice of baseline, the experimental setup seems well done.
3. The analysis for Class Conditional Coverage seems quite thorough.

**Weaknesses:**

1. The abstract is inappropriately long. The in-depth descriptions of the contributions should be reserved for the introduction to improve readability.
2. The notation of the paper is slightly confusing. For example, you use $r_f(X, Y)$ at the beginning of Section 4.2, where $X, Y$ are datasets. However, earlier, you use $r_f(X_{n+1}, y)$ correspond to a specific datapoint $X_{n+1}, y$. Which is it? Moreover, one uses $\epsilon_y$ to denote the top-k error but $\epsilon_{n_y}$ to denote a general constant. Using $\epsilon$ for a general constant definition is not a great practice, in my opinion, and greatly complicates the reading.
3. The computation of $\hat{k}(y)$ that achieves sufficiently small $\epsilon_{y}$ seems very difficult. Indeed, the equation provided to calculate this calibrated class value seems very computationally expensive to find. Moreover, it seems that the quality of the generated sets is highly dependent on $g$. I believe two additional explorations should be included in this work. The first is an ablation oon how different values of $g$ affect the generated length of the sets. If the prediction sets' quality is only strong for different $g$, which vary greatly between different hyperparameters, then doing a hyperparameter search for different $g$ for every individual dataset is very expensive. Comparing a method for which the hyperparameters have been extensively tuned to another algorithm for which this is not the case is not a fair empirical comparison. Moreover, a detailed section on the computability of the calibrated class is in order. It seems that you need to do a linear search over the calibrated dataset many times, which is far more expensive than traditional Split Conformal Prediction. A breakdown here would be nice.
4. The assumption in Theorem 2, i.e., equation 10, is a very strong assumption. More proper detail should go into detailing when this assumption holds and how. It seems to me that the heavy lifting of the proof of this theorem is done by this strong assumption. It is not clear to me when this assumption should hold.
5.  I am surprised there is only one baseline for CCP. I believe a vast array of methods for CCP should be compared. This could include the methods from ("Class-Conditional Conformal Prediction With Many Classes", "Classification with Valid and Adaptive Coverage", "Improving Conditional Coverage via Orthogonal
Quantile Regression", "A Gentle Introduction to Conformal Prediction and Distribution-Free Uncertainty Quantification"). There is a vast array of CCP works, and comparing to the most basic method is insufficient, in my opinion.

**Questions:**

1. This notion of class conditional coverage is strictly weaker than the traditional conditional coverage often discussed in literature. Do existing conditional coverage methods such as those mentioned in "A Gentle Introduction to Conformal Prediction and
Distribution-free uncertainty Quantification" achieves suboptimal set length compared to this weaker notion of conditional coverage.
2. If the nominated coverage $\tilde{\alpha}$ is larger than $\alpha$, then why are the generated sets shorter? This is not immediately clear to me. Is it because this allows for the use of only the top-k classes, which shortens the intervals?
3. In the experiments, how were the hyperparameters chosen? I don't see where values such as $g$ were chosen. Were the hyperparameters for the baselines chosen in a similar manner to that of the proposed method?
4. Do existing works guarantee that their method of CCP outperforms the naive CCP method in terms of the size of the prediction set, or is this unique to this work?

---

> ### Author Response · Authors · 2023-11-19
> **Response For Reviewer KjCS**
>
> We thank the reviewer for their thoughtful comments and feedback.
>
> Q1: **The proof demonstrating superiority to naive CCP relies on a strong assumption.**
>
> A1: Theorem 2 is a general result and depends on very mild assumptions.
>
> 1. No assumption on top-$k$ accuracy
>
> We clarify that our $k$-CCP algorithm is model-agnostic, since it does not impose any assumed properties of the learned classifier and can be used with any pre-trained classifier. The only extra information (compared with standard CP, such as MCP and CCP) is to estimate the top-$k$ accuracy (or equivalently, top-$k$ error in Remark 2) to determine the calibrated rank $\widehat k(y)$. However, estimation of the top-$k$ error does not make any assumption on the learned classifier, since taking different values of $\widehat k(y)$ from $1$ to $C$ gives a full span of $[0, 1]$ for top-$k$ accuracy. Therefore, our method remains valid and practically classifier agnostic.
>
>
> 2. Assumption in Equation (10) (Equation (12) in our updated paper)  is commonly satisfied
>
> The only assumption of condition numbers $\sigma_y$ in Equation (12) is commonly satisfied and supported by empirical results (see Section 5.2, Figure 1, the last column with the standard training process). In addition, in Figure 24 of the updated paper, we also verify this assumption firmly holds on the calibration set.
>
> Q2: **The notation of the paper is slightly confusing. Using $r_f(X,Y)$, where $X,Y$ are datasets. However, earlier, using $r_f(X_{n+1},y)$ correspond to a specific datapoint $X_{n+1},y$. Which is it?**
>
> A2: We summarize our main notations in Section 2. Specifically, “$(X, Y)$ is a data sample”, and $\mathcal D$ denotes the dataset, e.g., the training set $\mathcal D_{\text{tr}}$ and the calibration set $\mathcal D_{\text{cal}}$. Moreover, $r_f(X, Y)$ denotes the rank of the ground-truth label $Y$ predicted by the classifier $f$ on input $X$.
>
> In Equation (3) of Section 3, we introduce “a new testing input $X_{n+1}$”. Accordingly, in Section 4.1 and 4.2, $r_f(X_{n+1}, y)$ denotes the rank of $y$ (this $y$ may be any label rather than the ground-truth one) predicted by classifier $f$ on the testing data $X_{n+1}$, where $y$ is a fixed label. Both CCP and $k$-CCP require iterating over all possible $y$ from $\mathcal Y$ (in our multiclass case, $\mathcal Y = \{1, \cdots,C\}$ for a total of $C$ classes).
>
> Q3: **The computation of $\hat k(y)$ that achieves sufficiently small ϵy seems very difficult. Indeed, the equation provided to calculate this calibrated class value seems very computationally expensive to find.**
>
> A3: Verifying if Equation 10 is satisfied in practice does not require significant computation/data overhead. The complexity for verification is $O(n C)$ (or improved to $O(n \log(C))$ with binary search over classes from top-$1$ to top-$C$ for a large value $C$) with $n$ calibration samples (or $n$ testing samples if verification on testing data) and $C$ classes. Under the standard i.i.d. condition, the empirical estimates approximate population values in $O(1/\sqrt{n})$.
>
> Q4: **Comparing a method for which the hyperparameters have been extensively tuned to another algorithm for which this is not the case is not a fair empirical comparison.**
>
> A4: Our $k$-CCP does not require more hyper-parameters to be tuned compared with baselines CCP and cluster-CP [R13] (an added baseline in the updated paper, Table 1, 15, 16, and 17, Figure 1, 2, 4, 5, and 24 in our updated paper). The only hyper-parameter $g$ has to be tuned for each CP method to achieve class-conditional coverage (their UCR close to $0$), under which we can compare their prediction set sizes as an efficiency measure. As a result, we believe our experiments provide a fair comparison.
>
> Q5: **Moreover, a detailed section on the computability of the calibrated class is in order.**
>
> A5: See A3 above Q3.
>
> Q6: **The assumption in Theorem 2, i.e., equation 10 (Equation (12) in our updated paper), is a very strong assumption.**
>
> A6: See A1 above Q1.
>
> Q7: **Do existing conditional coverage methods [12]  achieves suboptimal set length compared to this weaker notion of conditional coverage.**
>
> A7: Indeed the class-conditional coverage is a special notion of general conditional coverage (such as the group-conditional coverage of Mondrian conformal predictor [R18], [R13]). However, the general conditional CP can be realized to multiple subtypes, including attribute conformal prediction [R18] and input-conditional conformal prediction [R19]. Consequently, these different realized algorithms for various conditional coverages are not directly comparable.
>
> In our experiments, we have selected the CCP which is introduced in [R16] (also refer to [R17]) as the baseline, without comparing with other non-comparable conditional CP methods. In addition, we added a new baseline cluster-CP [R13], a CP method that achieves class-conditional coverage under some conditions over the learned clustering function.

---

> ### Author Response · Authors · 2023-11-19
>
> Q8: **If the nominated coverage $\tilde \alpha$  is larger than $\alpha$, then why are the generated sets shorter?  Is it because this allows for the use of only the top-$k$ classes, which shortens the intervals?**
>
> A8: Yes, the additional calibration for the top-$k$ class improves efficiency (produced smaller prediction sets). The $k$-CCP algorithm adopts the double calibration mechanism to achieve improved trade-off between coverage and prediction set size. The key advantage of the additional calibration over rank is to exclude those data where their true labels are ranked beyond top-$\widehat k(y)$ positions (i.e., poorly predicted data) in the calibration process, which in turn helps remove labels beyond top-$\widehat k(y)$ positions in the testing process.
>
>
> Q9: **In the experiments, how were the hyperparameters chosen? Were the hyperparameters for the baselines chosen in a similar manner to that of the proposed method?**
>
> A9: See A4 above Q4.
>
> Q10: **Do existing works guarantee that their method of CCP outperforms the naive CCP method in terms of the size of the prediction set, or is this unique to this work?**
>
> A10: We add a baseline cluster-CP [R13] that achieves approximate class-conditional coverage (see Table1, 15, 16, and 17 in the updated paper) and find our $k$-CCP out performs it in efficiency. In [R13], the authors empirically showed improved efficiency, but did not provide theoretical analysis or guarantee. In contrast, our paper theoretically analyzes the factors ($\sigma_y$ in Theorem 2) that reduce prediction set sizes of our $k$-CCP and also verifies in experiments in the last column of Figure 1 (we further verify these condition numbers on calibration data in Figure 24 of the updated paper).
>
> Q11: **Only compare the method to the simplest baseline of CCP.**
>
> A11: We added the following experimental results as requested by the reviewers.
>
> (1) Comparison with Cluster-CP [R13], an accepted NeurIPS-2023 paper, using both APS and RAPS scoring functions
>
> (2) Comparison experiments for MCP [R14], CCP [R16], [R17], Cluster-CP, and $k$-CCP with both APS and RAPS scoring functions
>
> (3) Comparison experiments with conformal training to show the synergistic benefits of CP methods including $k$-CCP using a better classifier
>
> (4) Ablation study for hyper-parameter $g$ with CCP, Cluster-CP, and $k$-CCP
>
> (5) Verification of $\sigma_y$ on calibration data for different datasets
>
> Input space conditional CP methods pointed out by reviewers are not applicable for output space conditional CP (i.e., class-conditional CP). So we could not add such comparison experiments. We would like to point out that we included all the baselines used by the Cluster-CP paper which is accepted for publication at NeurIPS-2023.
>
> Q12: **An ablation study on how different values of $g$ affect the generated length of the sets.**
>
> A12: We add the ablation study to verify how hyper-parameter $g$ affects the performance of CCP, cluster-CP, and $k$-CCP. We use the under coverage ratio (UCR) and average prediction set size (APSS) as metrics (see Figure 2, 5, and 23 in the updated paper). According to the results, we observe that UCR and APSS of $k$-CCP are much smaller than CCP and cluster-CP with the same $g$ value. It verifies that $k$-CCP achieves a better trade-off between coverage and prediction set size.
>
> [R13] Ding, T., Angelopoulos, A. N., Bates, S., Jordan, M. I., & Tibshirani, R. J. (2023). Class-Conditional Conformal Prediction With Many Classes. arXiv preprint arXiv:2306.09335.
>
> [R14] Romano, Y., Sesia, M., & Candes, E. (2020). Classification with valid and adaptive coverage. Advances in Neural Information Processing Systems, 33, 3581-3591.
>
> [R15] Feldman, S., Bates, S., & Romano, Y. (2021). Improving conditional coverage via orthogonal quantile regression. Advances in neural information processing systems, 34, 2060-2071.
>
> [R16] Angelopoulos, A. N., & Bates, S. (2021). A gentle introduction to conformal prediction and distribution-free uncertainty quantification. arXiv preprint arXiv:2107.07511.
>
> [R17] Vladimir Vovk. Conditional validity of inductive conformal predictors. In Asian conference on machine learning, pages 475–490. PMLR, 2012
>
> [R18] Vovk, V., Gammerman, A., & Shafer, G. (2005). Algorithmic learning in a random world (Vol. 29). New York: Springer.
>
> [R19] Sesia, M., & Romano, Y. (2021). Conformal prediction using conditional histograms. Advances in Neural Information Processing Systems, 34, 6304-6315.

---

> > ### Comment · Reviewer_KjCS · 2023-11-20
> > **Response**
> >
> > I thank this author for their thoughtful response.
> >
> > $\sigma_y$ is strictly less than $1$, so the Condition from Equation 12 should always be satisfied. Is this not true?
> >
> > I find that most of my weaknesses have been addressed. The only weakness I have is that there should still be more baselines. The notes on computability of $\hat{k}$, the ablations of $g$, and the remarks on when the assumption from equation 12 is satisfied should be added to the camera-ready version. I would greatly appreciate more baselines from any of the works I mentioned, if possible. However, I feel comfortable increasing my score and believe the paper should be accepted.

---

> ### Author Response · Authors · 2023-11-21
>
> We thank you for your valuable and insightful review; we appreciate your increasing score.
>
> FQ1: **$\sigma_y$ is strictly less than $1$, so the Condition from Equation 12 should always be satisfied. Is this not true?**
>
> A1: Yes. The empirical verification results show a stronger condition ($\sigma_y$ is much smaller than 1 for all $y$, see the last column of Figure 1 and Figure 4 in our revised paper ), which ensures that the assumption in Theorem 2 (Equation (12) ) is always satisfied. Note that the reduction in prediction set size of $k$-CCP over CCP is proportional to how small the $\sigma_y$ values are, which are classifier-dependent.
>
> FQ2: **The only weakness I have is that there should still be more baselines.**
>
> A2: We highlight that our revised paper includes a comprehensive comparison with most class-conditional conformal prediction algorithms, such as Cluster-CP [R23] and [R25] based CCP as baselines. For example, all baselines compared in [R23] are denoted as ''MCP‘’ and ''CCP‘’ in our paper. Note that [R23] is a NeurIPS-2023 accepted paper and our paper has all the baselines included in that paper. We note that papers mentioned by reviewers on conditional CP methods for input space [R26, R27]are not applicable to output space, i.e., the class-conditional setting. We updated the related work in the revised paper.
>
> FQ3: **The notes on computability of $\widehat k$, the ablations of $g$, and the remarks on when the assumption from equation 12 is satisfied should be added to the camera-ready version.**
>
> A3: Thank you for your suggestions. We have added the computability of $\widehat k$, the condition explanation of Equation (12), the ablation study of $g$, and the overall experiments with a new baseline [R23] and new score function [R21] in our updated paper. The revised paper reflects the camera copy version with all the changes.
>
> [R23] Ding, T., Angelopoulos, A. N., Bates, S., Jordan, M. I., & Tibshirani, R. J. (2023). Class-Conditional Conformal Prediction With Many Classes. arXiv preprint arXiv:2306.09335. Accepted for NeurIPS-2023.
>
> [R24] Angelopoulos, A., Bates, S., Malik, J., & Jordan, M. I. (2020). Uncertainty sets for image classifiers using conformal prediction. arXiv preprint arXiv:2009.14193.
>
> [R25] Stutz, D., Cemgil, A. T., & Doucet, A. (2021). Learning optimal conformal classifiers. arXiv preprint arXiv:2110.09192.
>
> [R26] Isaac Gibbs, John J Cherian, and Emmanuel J Cand`es. Conformal prediction with conditional guarantees. arXiv preprint arXiv:2305.12616, 2023
>
> [R27] Shai Feldman, Stephen Bates, and Yaniv Romano. Improving conditional coverage via orthogonal quantile regression. Advances in neural information processing systems, 34:2060–2071, 2021.

---

### Official Review · Reviewer_v9Qu · 2023-10-31

**Soundness:** 3 good
**Presentation:** 3 good
**Contribution:** 3 good
**Rating:** 6
**Confidence:** 4

**Summary:**

The authors introduce a new approach, based on conformal prediction, to obtain prediction sets with a fixed set size in the setting of multiclass classification. While ordinary conformal prediction only gives control over the coverage of prediction sets by carefully choosing the significance level, this new method allows to choose the set size by selecting labels based on their rank statistics and appropriately modifying the significance level of the ordinary conformal predictor as to preserve the validity guarantee.

In a simple experiment they show how their method consistently outperforms standard marginal and class-conditional (Mondrian) conformal prediction on a selection of datasets for various class imbalance degrees.

**Strengths:**

•	The lack of efficiency guarantees in conformal prediction is a huge drawback for imbalanced data, the method introduced in this paper circumvents this problem.

•	Since the authors give an explicit procedure on how to modify the initial significance level, the method is almost as easy to implement as standard conformal prediction methods.

**Weaknesses:**

•	The notations are sometimes a bit convoluted, e.g. in Proposition 1. Moreover, the terminology is sometimes somewhat non-standard. CCP from the paper is often called Mondrian conformal prediction in the literature, and CCP itself is also used as an abbreviation for cross-conformal prediction.

•	A few minor errors:
- Equation (6) hay label y on the left-hand side but uses c on the right-hand side.
- In the proof of Lemma 1, in the equation with the Chernoff bound, a factor n is missing in the expression in the middle.

•	Proposition 1 feels a bit overly complex. Although technically sound, the content should be straightforward. E.g. if the coverage of two classes should average to 0.9 and one is larger than that, the other is going to be lower. This has also been discussed in a recent paper by Ding et al. "Class-conditional conformal prediction with many classes", Neurips 2023.

•	I cannot see what Theorem 2 contributes. The theorem more or less assumes that the new method is better (the sigma factors), so obviously it will be better on average.

**Questions:**

See above.

---

> ### Author Response · Authors · 2023-11-19
> **Response For Reviewer v9Qu**
>
> We thank the reviewer for their thoughtful comments and feedback.
>
> Q1: **The terminology is sometimes somewhat non-standard. CCP from the paper is often called Mondrian conformal prediction in the literature.**
>
> A1: Thank you for your suggestion, but our $k$-CCP is not a Mondrian conformal predictor. In fact, Mondrian conformal prediction (MCP) is a general procedure that encompasses many kinds of conditional conformal prediction [R11] [R12], including class conditional conformal prediction and group (attribute) conditional conformal prediction, etc. Our paper focuses on only class-conditional conformal prediction, instead of the general conditional coverage. Therefore, we use the terminology “class-conditional conformal prediction” to represent the relevant algorithms.
>
> Q2: **Proposition 1 feels a bit overly complex. Although technically sound, the content should be straightforward. This has also been discussed in a recent paper by Ding et al. "Class-conditional conformal prediction with many classes", Neurips 2023**
>
> A2: Proposition 1 is a general result to show the ease of under-coverage for Marginal CP. Technically, we only use very mild assumptions in Equation (7) as the condition for the violation of class-conditional coverage. Although other literature may discuss the same observation, our Proposition 1 theoretically quantifies the gap of class-conditional coverage and importantly makes this quantity easily verifiable in experiments (See the first column of Figure 1 and 4).
>
> Q3: **What Theorem 2 contributes. The theorem more or less assumes that the new method is better (the sigma factors), so obviously it will be better on average.**
>
> A3: The contribution of Theorem 2 is to theoretically identify the source of reduced prediction set sizes by $k$-CCP, so that it enables verification in experimental results shown in Figure 1 (last column). We additionally verify it on calibration data in Figure 24 of the updated paper. We believe that Theorem 2 reveals this insight/knowledge and wraps up the key factors into the condition numbers $\sigma_y$.
>
>
>
> [R11] Vovk, V., Gammerman, A., & Shafer, G. (2005). Algorithmic learning in a random world (Vol. 29). New York: Springer.
>
> [R12] Ding, T., Angelopoulos, A. N., Bates, S., Jordan, M. I., & Tibshirani, R. J. (2023). Class-Conditional Conformal Prediction With Many Classes. arXiv preprint arXiv:2306.09335.

---

### Official Review · Reviewer_RD2z · 2023-11-01

**Soundness:** 3 good
**Presentation:** 3 good
**Contribution:** 2 fair
**Rating:** 5
**Confidence:** 3

**Summary:**

The paper theoretically proves that Marginal Conformal Prediction can result in over- or under-coverage for classes in imbalanced data settings. To address this, the authors propose the k-Class-Conditional Conformal Prediction (k-CCP) approach, which integrates inflated coverage with a calibrated rank threshold, derived from the top-k error of the classifier for each class. Supported by both theoretical proofs and experiments on various datasets, the authors shows k-CCP outperforms CCP on average prediction set size.

**Strengths:**

- The paper addresses the issue of imbalanced data under the conformal prediction framework, and provides a feasible way of optimizing class-specific coverage while achieving shorter prediction intervals. Both theoretical proofs and empirical evidence support the efficacy of the proposed method.
- The k-CCP algorithm is a novel approach that combines the strengths of CCP with additional refinements to handle imbalanced data more effectively.
- Overall, the paper is well-written and consistent.

**Weaknesses:**

- The purposed method heavily relies on the ranking of candidate class labels by the classifier, which could be a limitation if the classifier's ranking is not reliable.
- Chapter 3 appears somewhat redundant. Using an overall score for each class in conformal prediction (MCP) would lead to marginal coverage rather than class-specific coverage seems evident. Previous works by (Lei, 2014; Sadinle et al., 2019) studied overall coverage and class-specific coverage separately.
- The empirical results could benefit from further enhancement (see questions below).

**Questions:**

- In Theorem 2, could assumption (10) be too strong to achieve? It suggests that kCCP algorithm should lead to a coverage that is less than or equal to the CCP and achieve a shorter length as a trade-off. Also, could you elaborate on how this assumption was validated?

- I might have overlooked this detail, but how can we choose $k$ by hyper-parameter $g$ in general?

- The Average Prediction Set Size (APSS) you presented takes the average prediction length evenly for each class. I wonder if you can present the comparison of the overall average prediction lengths. I'm curious if we will have a larger set-valued prediction for the ''majority'' classes.

---

> ### Author Response · Authors · 2023-11-19
> **Response For Reviewer RD2z**
>
> We thank the reviewer for their thoughtful comments and feedback.
>
> Q1: **The purposed method heavily relies on the ranking of candidate class labels by the classifier, which could be a limitation if the classifier's ranking is not reliable.**
>
> A1: No, $k$-CCP does not make assumptions on the top-$k$ accuracy to configure (see Theorem 1 and Remark 2).
>
> $k$-CCP degenerates to CCP in the worst-case when the label rank threshold for each class is set to the total number of classes. Intuitively, the reduction in prediction set sizes from $k$-CCP over CCP depends on how small the label rank thresholds are which is classifier-dependent. If the top-$k$ accuracy of the classifier is good, then we will achieve significantly small prediction sets using $k$-CCP (Theorem 2 and condition numbers $\sigma_y$). Our experimental results on four diverse datasets demonstrate that $k$-CCP significantly reduces prediction set size over CCP and Cluster-CP (NeurIPS-2023 paper).
>
>
>
> Q2: **Chapter 3 appears somewhat redundant. Using an overall score for each class in conformal prediction (MCP) would lead to marginal coverage rather than class-specific coverage seems evident. Previous works by (Lei, 2014; Sadinle et al., 2019) studied overall coverage and class-specific coverage separately.**
>
> A2: We do not use the fact that MCP does not guarantee class-conditional coverage as the only key motivation. Instead, we use its failure analysis to highlight the ease of under-coverage on imbalanced data (see the first column of Figure 1 for significant deviation of class-wise quantiles even for relatively balanced data where $\rho = 0.5$). On the other hand, the existing CCP method guarantees the class-conditional coverage but produces large prediction sets. These two observations together motivate the need for efficient CP algorithms which provide class-conditional coverage with small prediction sets.
>
> Q3: **In Theorem 2, could assumption (10) (Equation (12) in our updated paper) be too strong to achieve?**
>
> A3: The assumption of condition numbers $\sigma_y$ in Equation (12) is commonly satisfied and supported by empirical results (see Section 5.2, Figure 1, the last column with the standard training process). In addition, in Figure 24 of the updated paper, we also verify this assumption firmly holds on the calibration set.
>
> Q4: **How this assumption was validated?**
>
> A4: To verify the assumption in Equation (10), we directly estimate the empirical values of $\sigma_y$ for each class label $y$ on testing data (Section 5.2, the last column of Figure 1). For fixed $y$, we go over all testing samples $X_{n+1}$ and calculate the frequencies of two events:
> (1) the joint event “the coverage of non-conformity scores and calibrated rank”, i.e., $V(X_{n+1}, y) \leq \widehat Q_{1-\tilde \alpha}^{\text{class}}$ with $\tilde \alpha \leq \alpha$” and meanwhile the rank of $y$ stays in top-$\widehat k(y)$, i.e., $r_f(X_{n+1}, y) \leq \widehat k(y)$
>
> (2) the event “the coverage of non-conformity scores”, i.e., $V(X_{n+1}, y) \leq \widehat Q_{1-\alpha}^{\text{class}}$ with nominated mis-coverage $\alpha$”.
>
> After that, we compute the fraction of these two frequencies, to estimate $\sigma_y$.
>
>
> Q5: **How can we choose $k$ by hyper-parameter $g$ in general?**
>
> A5: In Appendix C.2, we explained that we selected the value of $g$ from the interval $[0.1, 1]$ in increments of $0.05$ to find the minimal $g$ that $k$-CCP and CCP achieved the target class conditional coverage. We will move this explanation to the main text in the revised paper.
>
> Q6: **Present the comparison of the overall average prediction lengths.**
>
> A6: We report the overall average prediction lengths on the balanced testing set.
> Because our goal is class-conditional coverage, the macro average prediction set size is more suitable for fair comparison. Additionally, we use a balanced calibration dataset, even though the training dataset itself is imbalanced. It means that the macro average prediction set size equals the micro average prediction set size, which is also adapted by another class conditional conformal prediction method [R9].

---

> ### Author Response · Authors · 2023-11-19
>
> Q7: **If we will have a larger set-valued prediction for the ''majority'' classes.**
>
> A7: We added new experiments to compare the average prediction set size of majority, medium, and minority groups in MCP, CCP, cluster_CP [R9], and  $k$-CCP (See Figure 21 with APS scoring function and Figure 22 with RAPS [R10] scoring function in Appendix H of the revised paper).  We select the top $1/4$ classes of the largest number of data to the majority group. Similarly, we assign the bottom $1/4$ classes of smallest number of data to the minority group and the remaining $1/2$ classes to the medium group. In conclusion, for the same CP method, we find that the group-wise average prediction set sizes do not differ too much, while the sizes generated by different CP methods can be large, e.g., $k$-CCP outperforms baselines significantly.
>
> [R9] Ding, T., Angelopoulos, A. N., Bates, S., Jordan, M. I., & Tibshirani, R. J. (2023). Class-Conditional Conformal Prediction With Many Classes. arXiv preprint arXiv:2306.09335.
>
> [R10] Angelopoulos, A., Bates, S., Malik, J., & Jordan, M. I. (2020). Uncertainty sets for image classifiers using conformal prediction. arXiv preprint arXiv:2009.14193.

---

> ### Comment · Reviewer_RD2z · 2023-11-22
>
> Thank you for your detailed responses which I have carefully reviewed. While they haven't altered my initial assessment of the paper, I appreciate the clarity and effort put into addressing the concerns raised.

---

### Official Review · Reviewer_TQMX · 2023-11-01

**Soundness:** 2 fair
**Presentation:** 1 poor
**Contribution:** 2 fair
**Rating:** 3
**Confidence:** 4

**Summary:**

The authors propose a method to construct prediction sets using conformal prediction and the top-k predictions of a classifier. More specifically they chose to include in the prediction sets labels that satisfy the following a) have a conformal score below an class conditional quantile for a higher coverage than the target one, and b) have a classifier score ranking below a chosen threshold.
The objective of the proposed method is to achieve class conditional coverage guarantees and smaller size of the prediction sets. The authors evaluate their method on several datasets against conformal prediction with marginal guarantees and class conditional conformal prediction.

**Strengths:**

Imbalanced datasets are a significant problem that can have catastrophic consequences in the performance of predictive ML models. In this context, studying the problem of conformal prediction and aiming at robust methods with rigorous coverage guarantees are of great interest.

The experimental evaluation is on several datasets assuming different scenarios on the class imbalance during training.

The authors provide the code for reproducibility.

**Weaknesses:**

**Significance**

“Our proposed k-CCP algorithm (summarized in Algorithm 2) avoids scanning all class labels y ∈ Y by leveraging good properties of the given classifier f in terms of its top-k accuracy.” One of the advantages of conformal guarantees is that they do not depend on the classifier used to compute the prediction sets. Assumptions on the classifier make the method no longer applicable as a post-hoc procedure working with a black-box ML model. In addition,  in realistic scenarios it  may be hard to make valid assumptions on the top-k  accuracy of the classifier. Moreover, if a classifier is trained using an imbalanced dataset, one should expect the accuracy of the classifier to be higher for majority classes, which does not necessarily guarantee good performance on the top-$k$ accuracy.

The computation of $\hat{k}$ depends on the hyper parameter $g$. One would expect that tuning such a hyper parameter would add a data and computation overhead. The authors do not elaborate on this overhead neither on the process of selecting  the hyper parameter as well as the implications that this process can have. Furthermore, it is not clear if the proposed $\hat{k}$ will never be such that it violates the target coverage $1-\alpha$. A definition of $\hat{k}$ in theorem 1 as well as guarantees that the error terms in theorem 1 are small or can be small with high probability for that $\hat{k}$  could strengthen the contribution.

The results of theorem 2 strongly depend on the assumption (10). This assumption seems to depend on the probability distributions that may be hard to safely evaluate and verify in practice without a significant data overhead. It would be very useful to provide insights, or data efficient method with which one could check if (10) is satisfied.

**Novelty**

If the paper was the first attempt towards class conditional coverage, a failure analysis for the standard split conformal prediction algorithm, would make perhaps more sense. However, given that there are already other algorithms to address class conditional coverage, showing that MCP does not necessarily provide class-conditional guarantees does not seem as a strong motivation  for the current work.  Existing prior work [2] has already provide evidence that MCP does not satisfy class conditional guarantees. Besides MCP was never designed or claimed to achieve class conditional coverage.

Given that in the experimental results k-CPP and CPP seem to perform very similarly in terms of conditional coverage, it seems that the main advantage of k-CPP is the reduced set size. However, there is no comparison with CPP using the RAPS method [1] that improves over APS. It would be also important to investigate (theoretically and/or experimentally) how the proposed method compares to other approaches with conditional guarantees such as [3], or other approaches to improve the size of the prediction sets [4]. Were there results on how the proposed approach advances over such works, would make the contribution much stronger.

**Presentation/clarity**

The entire manuscript gives the reader the impression that it was written in a rush as, typos, grammatical errors and poor sentence structure appear very frequently.  These make the work cumbersome to read. Moreover, there seems to be quite some room fro improvement in terms of clarity, flow and cohesion. For example:
*   In the abstract it is very unclear what the authors mean by “inflated coverage and calibrated rank thresholds”
*   In the introduction in the 3rd paragraph in the middle  “To answer the main research question,”  should follow right after the main research question for a nice flow
*   In the last paragraph before contributions the way the authors are presenting the differences is quite confusing. Instead of stating the methods followed by CCP and then listing the methods of k-CCP, it would have been clearer to compare the two approaches point by point.
*  The three bullets in the contribution are hard to read. They lack of proper sentence structure, especially the lack of any articles make the reader struggling to follow. They look more like incomplete notes, than proper text.
*  The definition of the top-k error for a class $c$ is confusing. $\epsilon_c$ seems to depend both on the constants $c$ and $k$. Perhaps using $\epsilon_{c}^{k}$ could make the definition clearer.
*  In page 5, in the definition of the top-k error it the random variable $Z$ is not defined. Also, in the same paragraph it is not clear what is the distinction between $\hat{k}(y)$ and $k(y)$. Also $\hat{k}(y)$ is not formally defined the first time that it is introduced, which is confusing.
*  It is not clear why $\sigma_{y}$ is necessary in (10), as the denominator cancels out. One could directly state the condition in (10) using just the nominator of $\sigma_{y}$.

In the experimental evaluation, in table 1 the results for the Food-101 datasets on the UCR are not highlighted for CPP method wherever it achieves the lowest UCR among the baselines.

The under coverage indicator in the Under Coverage Ratio  assumes a fixed calibration set. However, conformal coverage guarantees on $1 - \alpha$ are in expectation over the calibration set and the test sample. As a result the UCR definition appears incorrect.

**Typos/Misc**

1. Abstract, 1st line “Classification tasks … arises” —> “Classification tasks…. arise”
2. Abstract, 8th line from the end, “estimates class-specific non conformity score threshold, inflated coverage and calibrated rank threshold”—> , “estimates class-specific non conformity score thresholds inflated coverage and calibrated rank thresholds”
2. Introduction, 2nd line of 1st paragraph, “with long tail distribution” —> “with long tail distributions"
3. “…minority classes are typically very important”; that seems a bit as an over-generalization. Depending on the setting there might be minority classes that are not necessarily crucial. Yet, it is true that it can happen that the minority classes are very important. “…minority classes can be very important..” might be a better phrasing.
4. Introduction, first paragraph, one line before the  end “imbalance data”—>  “imbalanced data”
6. Introduction,  in the paragraph before the contributions : “2) calibrated rank threshold for each class c.”—> “2) a calibrated rank threshold for each class c.“
7. Introduction, contributions “Novel CP algorithm for class-conditional coverage by calibrating conformity score and rank threshold pair for each class to exploit the top-k accuracy of the given classifier. “ —> “A novel CP…calibrating conformity scores/ the conformity score”. It is unclear what is a "rank threshold pair? Would it be better perhaps to write “by calibrating a pair of thresholds, one based on the conformity score, one based on the ranking of the score of the classifier”?
9. Introduction, contributions “Theoretical analysis to demonstrate the failure of marginal CP, k-CCP guarantees class- conditional coverage, and k-CCP produces smaller prediction sets over the baseline CCP.”—> “Theoretical analysis to demonstrate the failure of marginal CP, to prove class-conditional coverage guarantees of k-CCP, as well as to prove that k-CPP achieves to produce smaller prediction sets than CPP.”
10. Section 2, 1st paragraph 4th line “We consider imbalanced data setting” —> “We consider imbalanced data settings/ an imbalanced data setting”
11. Section 2, 1st paragraph 6th line “the soft classifier” —> “a soft classifier”
12. Section 2, in paragraph Problem definition “for imbalanced data setting” —> “for imbalanced data settings”
13. Above (1) “pupulation level” —> “population level”
14. Section 3, MCP paragraph line 3, “large value means that” —> “if the non-conformity score is large, the new…”
15. Section 3, MCP paragraph line 4 “conforms less with calibration samples” —> “conforms less to the calibration samples”
16. Section 3 Failure Analysis of MCP. “define class-wise empirical quantile” —> “define the class-wise empirical quantile”
17. Page 9 in paragraph above “conformal prediction” and above “summary”: “for imbalanced data setting” —> “for imbalanced data settings”
18. Page 9, conclusion “Our theoretical and empirical analysis demonstrate that MCP algorithm achieves only marginal coverage can arbitrarily have over- or under-coverage on classes in practice” —> is an “and” missing between “coverage” and “can”?
19. Page 9, conclusion “that estimates class-specific non-conformity score threshold” —> “that estimates class-specific non-conformity score thresholds”
20. Page 9, conclusion “calibrated rank” is unclear.
21. Page 9, conclusion “which satisfy both class-specific threshold and calibrated rank to produce the prediction set, in contrast with CCP baseline that iterates all possible class labels with non-conformity score threshold only” does not make sense.
22. Page 9, conclusion ”k-CCP produce” —> “k-CCP produces”
23 The table captions do not follow the ICRL author instructions.

[1] Angelopoulos, A., Bates, S., Malik, J., & Jordan, M. I. (2020). Uncertainty sets for image classifiers using conformal prediction. arXiv preprint arXiv:2009.14193.

[2] Löfström, T., Boström, H., Linusson, H., & Johansson, U. (2015). Bias reduction through conditional conformal prediction. Intelligent Data Analysis, 19(6), 1355-1375.

[3] Gibbs, I., Cherian, J. J., & Candès, E. J. (2023). Conformal Prediction With Conditional Guarantees. arXiv preprint arXiv:2305.12616.

[4] Stutz, D., Cemgil, A. T., & Doucet, A. (2021). Learning optimal conformal classifiers. arXiv preprint arXiv:2110.09192.

**Questions:**

1.In the definition of $\epsilon_{c}$ is the probability over the samples X and the classes Y? It might be helpful to clarify that in the problem definition.

2. In eq. 1 the authors define their class-conditional coverage objective. This objective includes the coverage probability on a population level. However the authors do not specify if the probability  is both over the calibration set and the test set as it is for CP. It would be helpful to clearly state over what is the coverage probability in eq. 1

3. Below eq 3. “good non-conformity scoring functions”, this term is rather vague for non experts in conformal prediction. It would have been clear if the authors mention what is a good non-conformity scoring function, (e.g., one that results in small prediction sets). The comment applies for the next line “effective conformity score function”. It is unclear what is effective, and if by effective, again the authors mean, resulting in smaller prediction set size.

---

> ### Author Response · Authors · 2023-11-19
> **Response For Reviewer TQMX**
>
> We thank the reviewer for their thoughtful comments and feedback.
>
> Q1: **Assumptions on the classifier make the method no longer applicable as a post-hoc procedure working with a black-box ML model. In addition, in realistic scenarios it may be hard to make valid assumptions on the top-k accuracy of the classifier.**
>
> A1: No, $k$-CCP does not make assumptions on the top-$k$ accuracy to configure (see Theorem 1 and Remark 2).
>
> $k$-CCP degenerates to CCP in the worst-case when the label rank threshold for each class is set to the total number of classes. Intuitively, the reduction in prediction set sizes from $k$-CCP over CCP depends on how small the label rank thresholds are which is classifier-dependent. If the top-$k$ accuracy of the classifier is good, then we will achieve significantly small prediction sets using $k$-CCP (Theorem 2 and condition numbers $\sigma_y$). Our experimental results on four diverse datasets demonstrate that $k$-CCP significantly reduces prediction set size over CCP and Cluster-CP (NeurIPS-2023 paper).
>
>
> Q2:  **The authors do not elaborate on this overhead neither on the process of selecting the hyper parameter as well as the implications that this process can have.**
>
> A2: Tuning the hyper-parameter $g$, similar to the model selection process for the other hyper-parameters like learning rate, only requires minor computational overhead. We select the $g$ from the interval [0.1, 1] to find the minimal $g$ that UCR of $k$-CCP and CCP equal to $0$. The detailed tuning approach is introduced in Remark 2 and Appendix C.2. Moreover, it is necessary for every CP method (including compared baselines) to tune the effective $1-\alpha$ to ensure their UCR is smaller than a universal threshold close to $0$ in practical experiments (refer to our response to Q12 for why UCR has to be set near $0$ for all class-conditional CP methods uniformly). In other words, tuning this hyperparameter $g$ for $k$-CCP does not cost more overhead compared to baselines.
>
> Q3: **It is not clear if the proposed $\hat k$ will never be such that it violates the target coverage $1−\alpha$.**
>
> A3: Under the standard i.i.d. condition, the estimated top-$k$ error concentrates around the population value. Although a concentration error occurs, it is in $O(1/\sqrt{n})$ and can be merged with the tuning process of the hyper-parameter $g$ for estimating the quantile $Q_{1-\alpha}^{\text{class}}$ (their concentration errors are in the same order). As a result, the calibrated $\widehat k(y)$ can be accurate enough to achieve the target class-conditional coverage as demonstrated by our strong empirical results across multiple benchmark datasets.
>
>
> Q4: **A definition of $\hat k$ in theorem 1 as well as guarantees that the error terms in theorem 1 are small or can be small with high probability for that $\hat k$ could strengthen the contribution.**
>
> A4: The notion of $\widehat k(y)$ is given in Equation (9) and the paragraph before Theorem 1 provides this detail, which says ``can be any integer from $1$ to $C$‘’. We do not determine the value of $\widehat k(y)$ because we would like Theorem 1 to be general. In Remark 2, we give how to determine their values from the feasible set in a unified framework that meanwhile handles the constant $g$ from concentration errors.
> For the top-$k$ errors ($\epsilon_y$ in our paper), we do not assume they are very small. Instead, we select the value $k$ such that the corresponding top-$k$ error stays in a certain range, i.e., $\leq \alpha - g/\sqrt{n_y}$. Among these feasible values, we pick the minimal one, i.e., determining $\widehat k(y)$ as per Remark 2.
> As for the concentration errors ($\varepsilon_{n_y}$ in our paper), it depends on the number of class-wise calibration samples $n_y$.
>
>
> Q5: **This assumption seems to depend on the probability distributions that may be hard to safely evaluate and verify in practice without a significant data overhead. It would be very useful to provide insights, or data efficient method with which one could check if (10) is satisfied.**
>
> A5: Verifying if Equation 10 is satisfied in practice does not require significant computation/data overhead. The complexity for verification is $O(n C)$ (or improved to $O(n \log(C))$ with binary search over classes from top-$1$ to top-$C$ for a large value $C$) with $n$ calibration samples (or $n$ testing samples if verification on testing data) and $C$ classes. Under the standard i.i.d. condition, the empirical estimates approximate population values in $O(1/\sqrt{n})$.
>
> In our verification experiments, it reduces to verifying if condition numbers $\sigma_y \leq 1$. If calibration data and testing data are i.i.d., then this verification experiment can also be done on the calibration set (See Section I, Figure 24 in our updated paper).

---

> ### Author Response · Authors · 2023-11-19
>
> Q6: **Showing that MCP does not necessarily provide class-conditional guarantees does not seem as a strong motivation for the current work.**
>
> A6: We do not use the fact that MCP does not guarantee class-conditional coverage as the only key motivation. Instead, we use its failure analysis to highlight the ease of under-coverage on imbalanced data (see the first column of Figure 1 for significant deviation of class-wise quantiles even for relatively balanced data where $\rho = 0.5$). On the other hand, the existing CCP method guarantees the class-conditional coverage but produces large prediction sets. These two observations together motivate the need for efficient CP algorithms which provide class-conditional coverage with small prediction sets.
>
>
> Q7: **Typos, grammatical errors and presentation in text and captions in the paper.**
>
> A7: Thank you for your careful corrections and suggestions. We will fix typos, grammatical errors, unclear presentation and complicated notations in text/tables/captions in the revised paper.
>
> Q8: **In page 5, in the definition of the top-k error it the random variable $Z$ is not defined. Also, in the same paragraph $\hat k(y)$ is not formally defined and lack of the difference of $k(y)$ and $\hat k(y)$.**
>
> A8: We highlight that  $\widehat k(y)$ is formally defined in Equation (9) and the paragraph before Theorem 1. Later we introduce how to determine its value in Remark 2. However, we do not use $k(y)$ in our paper.
> For the notation $Z$, we define it as $Z = (X, Y)$ as the features-label pair. We will include a clear definition of the random variable $Z$ in revision.
>
> Q9: **It is not clear why $\sigma_y$ is necessary in (10), as the denominator cancels out.**
>
> A9: We use $\sigma_y$ for a simpler notion of the condition number of the learned classifier. Our ablation study verifies the validity of Theorem 2 (see Section 5.2, the last column of Figure 1, $\sigma_y$ is exactly the X-axis).
>
> Q10: **The under coverage indicator in the Under Coverage Ratio assumes a fixed calibration set. However, conformal coverage guarantees on $1−\alpha$ are in expectation over the calibration set and the test sample. As a result the UCR definition appears incorrect.**
>
> A10: We disagree that the UCR definition appears incorrect as a measure of under-coverage. Although CCP algorithms guarantee class-conditional coverage in expectation, we do not have access to the underlying distribution, so we can only measure the coverage empirically. As a result, we define UCR as the measure of under-coverage on a realized testing set, i.e., it is the percentage of classes on which under-coverage occurs. To better avoid the bias of the realized testing set, we also “repeat experiments over 10 different random calibration-testing splits and report the average performance with standard deviation”. Therefore, we believe UCR is a reasonable metric to measure the class-conditional coverage and provides a fair condition for efficiency (prediction set size) comparison over CP methods.
>
> Q11: **There is no comparison with CPP using the RAPS method [R4].**
>
> A11: We added the following experimental results as requested by the reviewers.
>
> (1) Comparison with Cluster-CP, an accepted NeurIPS-2023 paper, using both APS and RAPS scoring functions
>
> (2) Comparison experiments for MCP, CCP, Cluster-CP, and $k$-CCP with both APS and RAPS scoring functions
>
> (3) Comparison experiments with conformal training to show the synergistic benefits of CP methods including $k$-CCP using a better classifier
>
> (4) Ablation study for hyper-parameter $g$ with CCP, Cluster-CP, and $k$-CCP
>
> (5) Verification of $\sigma_y$ on calibration data for different datasets
>
> Input space conditional CP methods pointed out by reviewers are not applicable for output space conditional CP (i.e., class-conditional CP). So we could not add such comparison experiments. We would like to point out that we included all the baselines used by the Cluster-CP paper which is accepted for publication at NeurIPS-2023.

---

> ### Author Response · Authors · 2023-11-19
>
> Q12: **How the proposed method compares to other approaches with conditional guarantees such as [R5], or other approaches to improve the size of the prediction sets [R6].**
>
> A12: The target of [R5] is conditional coverage conditioned on the (input space) feature $X \in G$, i.e., $X$ belongs to “pre-specified groups”. Instead, our paper focuses on the class-conditional coverage that is conditioned on (output space) each class, i.e., $Y = y$. We believe these two coverage guarantees are not comparable.
> [R6] proposed a training algorithm that aims to improve the size of prediction sets constructed by the subsequent CP algorithms. In contrast, our $k$-CCP is a model-agnostic (not a training algorithm) class-conditional CP algorithm with smaller prediction sets. Consequently, [R6] and our $k$-CCP are not comparable. However, we can still compare the efficiency of our $k$-CCP with other baselines based on the model trained by the learning algorithm in [R6].
> We add these experimental results in Table 17 in the updated paper to compare $k$-CCP with CCP and cluster-CP [R6] based on two score functions (APS and RAPS [R4]) using the model trained by [R5]. Results still verify that our $k$-CCP significantly reduces the size of prediction sets.
>
>
> [R4] Angelopoulos, A., Bates, S., Malik, J., & Jordan, M. I. (2020). Uncertainty sets for image classifiers using conformal prediction. arXiv preprint arXiv:2009.14193.
>
> [R5] Gibbs, I., Cherian, J. J., & Candès, E. J. (2023). Conformal Prediction With Conditional Guarantees. arXiv preprint arXiv:2305.12616.
>
> [R6] Stutz, D., Cemgil, A. T., & Doucet, A. (2021). Learning optimal conformal classifiers. arXiv preprint arXiv:2110.09192.
>
> [R7] Ding, T., Angelopoulos, A. N., Bates, S., Jordan, M. I., & Tibshirani, R. J. (2023). Class-Conditional Conformal Prediction With Many Classes. arXiv preprint arXiv:2306.09335.
>
> [R8] Vladimir Vovk. Conditional validity of inductive conformal predictors. In Asian conference on machine learning, pages 475–490. PMLR, 2012

---

> > ### Comment · Reviewer_TQMX · 2023-11-22
> >
> > I would like to thank the authors for taking the time to make major revisions to improve the manuscript and to clarify several procedures and assumptions. The responses are greatly appreciated, but unfortunately have not fully addressed the initial concerns, thus I will have to retain my score. The revised version is quite improved and a resubmission would benefit the work.
> >
> > Re: A1 it is clear that the proposed method does not rely on classifier assumptions to achieve validity, i.e., prediction sets that satisfy the class conditional target  coverage. However, from the paper, as well as A1, it is clear that the efficiency of the prediction sets, i.e., how small is their size, *does* depend on the assumption that the classifier has good top-k accuracy “If the top-k accuracy of the classifier is good, then we will achieve significantly small prediction sets using k-CCP”. As achieving prediction sets of smaller size seems the main advantage of the proposed method, the above implies that having this advantage relies on the assumption that the classifier has good top-k accuracy.
> >
> > Re: A3 and A4 heuristically tuning $g$ and $\hat{k}$ seems to strongly depend on the granularity of the range of the candidate $g$ values, as well as the size of the label space $C$. As a result high granularity as well as a very large label space would imply higher overhead; a limitation that should be explicit. In addition, even though the empirical results show that $\hat{k}$ satisfies coverage in practice, it would be very interesting to provide theoretical guarantees that the proposed method in remark 2 will always (or with high probability) satisfy coverage.
> >
> > Re: A5 estimating the probabilities in (12) in the revised version (that is (10) in the original version)  would require enough data for faithful estimates from each different class. In addition, even if with enough data, each estimate would be close to the true value with some probability. Assuming that this event holds *simultaneously*  for the probability estimates for all classes would require applying union bound or some type of multiplicity correction, which is not obvious from (12). Nevertheless, it would be useful in addition to the experimental results showing that (12) holds in practice, to provide theoretical results, showing guarantees that one could safely verify (12) using estimators from data in practice.

---

### Author Response · Authors · 2023-11-19
**Response to All:**

We thank all reviewers for their thoughtful comments and constructive feedback. We have updated our paper, where revisions to the original content are colored blue (typos, grammar, and clarification). Additional experimental results requested by the reviewers are reported in Appendix D (Experiments with cluster-CP Using APS Score Function), E (Comparison Experiments Using RAPS Score Function), F (Comparison Experiments with Conformal Training Model), G (Illustration of group-wise Average Prediction Size), H (Ablation Study for hyper-parameter $g$) and I (Verification of $\sigma_y$ on calibration datasets).

Below are our responses to the common questions.

CQ1. **Does $k$-CCP assume any assumption for the top-k accuracy of the pre-trained model?**

A1: No. We clarify that our $k$-CCP algorithm is model-agnostic, since it does not impose any assumed properties of the learned classifier and can be used with any pre-trained classifier. The only extra information (compared with standard CP, such as MCP and CCP) is to estimate the top-$k$ accuracy (or equivalently, top-$k$ error in Remark 2) to determine the calibrated rank $\widehat k(y)$. However, estimation of the top-$k$ error does not make any assumption on the learned classifier, since taking different values of $\widehat k(y)$ from $1$ to $C$ gives a full span of $[0, 1]$ for top-$k$ accuracy. Therefore, our method remains valid and practically classifier agnostic.

CQ2. **The computation cost for verifying the assumption $\sigma_y$ in Theorem 2.**

A2: Verifying if Equation 10 is satisfied in practice does not require significant computation/data overhead. The complexity for verification is $O(n C)$ (or improved to $O(n \log(C))$ with binary search over classes from top-$1$ to top-$C$ for a large value $C$) with $n$ calibration samples (or $n$ testing samples if verification on testing data) and $C$ classes. Under the standard i.i.d. condition, the empirical estimates approximate population values in $O(1/\sqrt{n})$.

CQ3. **The reason to show that MCP does not necessarily provide class-conditional guarantees.**

A3: We do not use the fact that MCP does not guarantee class-conditional coverage as the only key motivation. Instead, we use its failure analysis to highlight the ease of under-coverage on imbalanced data (see Figure 1 for significant deviation of class-wise quantiles even for relatively balanced data where $\rho = 0.5$). On the other hand, the existing CCP method guarantees the class-conditional coverage but produces large prediction sets. These two observations together motivate the need for efficient CP algorithms which provide class-conditional coverage with small prediction sets.

CQ4.  **The reason to define Under Coverage Ratio (UCR) as a coverage metric.**

A4: Although CCP algorithms guarantee class-conditional coverage in expectation, we do not have access to the underlying distribution, so we can only measure the coverage empirically. As a result, we define UCR as the measure of under-coverage on a realized testing set, i.e., it is the percentage of classes on which under-coverage occurs. To better avoid the bias of the realized testing set, we also “repeat experiments over 10 different random calibration-testing splits and report the average performance with standard deviation”. Therefore, we believe UCR is a reasonable metric to measure the class-conditional coverage and provides a fair condition for efficiency (prediction set size) comparison over CP methods.

CQ5. **Is the assumption in Theorem 2 (Equation (12) in our updated paper) too strong?**

A5: The assumption of condition numbers $\sigma_y$ in Equation (12) is commonly satisfied and supported by empirical results (see Section 5.2, Figure 3 on all four datasets with the standard training process). In addition, in Figure 21 of the updated paper, we also verify this assumption firmly holds on the calibration set.

CQ6. **The contribution of Theorem 2.**

A6: The contribution of Theorem 2 is to theoretically identify the source of reduced prediction set sizes by $k$-CCP, so that it enables verification in experimental results shown in Figure 3. We additionally verify it on calibration data in Figure 21 of the updated paper. We believe that Theorem 2 reveals this insight/knowledge and wraps up the key factors into the condition numbers $\sigma_y$.

---

> ### Author Response · Authors · 2023-11-19
>
> CQ7: **Is there any baseline other than CCP [R1], [R2] for class conditional conformal prediction?**
>
> A7: We added an additional baseline cluster-CP [R3] that achieves approximate class-conditional coverage (see Table 15, 16, and 17 in the updated paper) and found that $k$-CCP outperforms it by producing significantly smaller prediction sets. In [R3], the authors empirically showed improved efficiency (small prediction sets), but did not provide theoretical analysis for exact class-conditional coverage. Moreover, their class-conditional coverage guarantee requires sufficiently good performance on the clustering map (see Proposition 3 therein). In contrast, our $k$-CCP does not. Instead, $k$-CCP is adaptive and classifier-agnostic (see our response to CQ1 above).
>
> CQ8. **Typos, grammatical errors, and presentation in text and captions in the paper.**
>
> A8: Thank you for your careful corrections and suggestions. We have fixed typos, grammatical errors, unclear presentation, and complicated notations in text/tables/captions in the revised paper and highlighted the new changes using blue color.
>
> [R1] Angelopoulos, A. N., & Bates, S. (2021). A gentle introduction to conformal prediction and distribution-free uncertainty quantification. arXiv preprint arXiv:2107.07511.
>
> [R2] Vladimir Vovk. Conditional validity of inductive conformal predictors. In Asian conference on machine learning, pages 475–490. PMLR, 2012
>
> [R3] Ding, T., Angelopoulos, A. N., Bates, S., Jordan, M. I., & Tibshirani, R. J. (2023). Class-Conditional Conformal Prediction With Many Classes. arXiv preprint arXiv:2306.09335.

---

### Author Response · Authors · 2023-11-21
**Summary of reviews and responses for the revised paper**

Dear Reviewers,

We thank all of you for asking good questions and providing critical feedback which greatly improved the revised paper.

We summarize the main concerns and how we addressed them below. We addressed all individual review comments in their respective responses. We revised the paper to reflect all these changes which are highlighted in blue color.

**1. Does $k$-CCP make any assumption for the top-$k$ accuracy of the classifier?**

No, $k$-CCP does not make assumptions on the top-$k$ accuracy to configure (see Theorem 1 and Remark 2).

$k$-CCP degenerates to CCP in the worst-case when the label rank threshold for each class is set to the total number of classes. Intuitively, the reduction in prediction set sizes from $k$-CCP over CCP depends on how small the label rank thresholds are which is classifier-dependent. If the top-$k$ accuracy of the classifier is good, then we will achieve significantly small prediction sets using $k$-CCP (Theorem 2 and condition numbers $\sigma_y$). Our experimental results on four diverse datasets demonstrate that $k$-CCP significantly reduces prediction set size over CCP and Cluster-CP (NeurIPS-2023 paper).

**2. Assumption in Theorem 2 too strong?**

The assumption of condition numbers $\sigma_y$ in Equation (12) of the revised paper is commonly satisfied and supported by empirical results (see Section 5.2, Figure 1 last column on mini-ImageNet and Food-101 datasets with the standard training process). In fact, $\sigma_y$ is much less than 1 in all cases which helps in reducing the prediction set size using $k$-CCP. In addition, in Figure 24 of the updated paper, we also verify this assumption firmly holds on the calibration set.

We can easily verify this condition with $n$  calibration samples (or testing samples if verification on testing data) and $C$ classes in $O(nlog(C))$.

**3. Reason to show the failure analysis of Marginal CP**

We do not use the fact that MCP does not guarantee class-conditional coverage as the only key motivation. Instead, we use its failure analysis to highlight the ease of under-coverage on imbalanced data (see Figure 1 for significant deviation of class-wise quantiles even for relatively balanced data where $\rho$=0.5). On the other hand, the existing CCP method guarantees the class-conditional coverage but produces large prediction sets. These two observations together motivate the need for efficient CP algorithms which provide class-conditional coverage with small prediction sets.

**4. Reason to define Under Coverage Ratio (UCR) as coverage metric**

Although CCP algorithms guarantee class-conditional coverage in expectation, we do not have access to the underlying distribution, so we can only measure the coverage empirically. As a result, we define UCR as the measure of under-coverage on a realized testing set, i.e., it is the percentage of classes on which under-coverage occurs. To better avoid the bias of the realized testing set, we also “repeat experiments over 10 different random calibration-testing splits and report the average performance with standard deviation”. Therefore, we believe UCR is a reasonable metric to measure the class-conditional coverage and provides a fair condition for efficiency (prediction set size) comparison over CP methods.

**5. Request for new experiments**

We added the following experimental results as requested by the reviewers.

(1) Comparison with Cluster-CP, an accepted NeurIPS-2023 paper, using both APS and RAPS scoring functions

(2) Comparison experiments for MCP, CCP, Cluster-CP, and $k$-CCP with both APS and RAPS scoring functions

(3) Comparison experiments with conformal training to show the synergistic benefits of CP methods including $k$-CCP using a better classifier

(4) Ablation study for hyper-parameter $g$ with CCP, Cluster-CP, and $k$-CCP

(5) Verification of $\sigma_y$ on calibration data for different datasets

Input space conditional CP methods pointed out by reviewers are not applicable for output space conditional CP (i.e., class-conditional CP). So we could not add such comparison experiments. We would like to point out that we included all the baselines used by the Cluster-CP paper which is accepted for publication at NeurIPS-2023.

**6. Main advantage of $k$-CCP over CCP and Cluster-CP**

The main research question of this paper is: how can we develop provable CP methods for the imbalanced data setting to produce small prediction sets?

$k$-CCP produces significantly smaller prediction sets compared to CCP and Cluster-CP on all datasets by achieving class-conditional coverage.

**7. Presentation and clarification**

We have incorporated all your feedback to improve both presentation and clarity in the revised paper. All changes are highlighted in blue color.

We hope our responses have addressed your concerns. We are happy to answer any followup questions.

Thanks,

Authors

---

### Meta-Review · Area_Chair_HP86 · 2023-12-08

**Metareview:**

Classification with imbalanced data is an important problem in ML and classifiers must be accompanied with a rigorous uncertainty quantification procedure. Conformal prediction is one such method that comes with strong theoretical average coverage guarantee.
Unfortunately, average coverage does not prevent undercoverage conditional on some class. This paper formally show that this occurs on underrepresented class, which can be problematic in medical diagnosis for example. A main contribution is to leverage top-k accuracy to design a conformal prediction method that achieve class conditional coverage along with a small prediction set size. An extensive numerical experiments illustrate the performance of the proposed methods.

Uncertainty quantification for imbalanced data is a very important topics in ML and the example showing inefficiency of a naive conformal methods is quite interesting. Simultaneously achieving class conditional coverage and small prediction sets is also highly relevant.

Unfortunately, the paper might have several clarity issues which make the arguments hard to follow. More specifically most of the assumptions seems hard to test, depends on the *unknown* ground-truth distribution, several hyperparameter to be tune which leverage further data dependence that can probably impact the coverage guarantee etc.

- The first point is to argue that the Marginal Conformal Prediction (MCP) lacks class conditional coverage. Ok but this seems to be well known and a simple modification, namely Mondrian Conformal Prediction https://arxiv.org/pdf/1209.2673.pdf achieves class-conditional coverage. Proposition 3 gives even a stronger results and weaker assumptions than what is proposed in this paper. This seems to be solved decades ago, and should be the minimal baseline. Some reviewers mentioned it and the responses of the authors seems quite unclear.

- The main result, Theorem 1, depends on a conditional error $\epsilon_y = \mathbb{P}_Z( r_f(Z) > \hat k(y) \mid Y = y ) $ for $Z = (X, Y)$ which is unknown since it depends on the ground-truth distribution. As such the choice of $\tilde \alpha_y$ is not safe ---as written in the paper ---. Several reviewers mentioned this and the authors replies that $\epsilon_y$ is an expectation and then can be approximated with its empirical estimate. The estimation error can be controlled via Hoeffding type bound which result in an extra term $O(1/\sqrt{n})$. This must be included in the tuning of $\tilde \alpha_y$ otherwise the result does not look correct. At least some more clarification are needed.

In all case, directly applying Mondrian Conformal Predictor to top-k accuracy (or any classifier that is symmetric wrt data permutation) automatically leads to class-conditional validity without any extra assumption. Again, this seems to be strictly stronger than the presented results. The authors seem aware of this but did not propose a transparent discussions in my opinion. Significant clarification are needed in order to assess the significance of this result.


- The theorem 1 requires tuning the parameter $g$ on the left-out data. Reviewers mentioned a potential overhead but the authors argue that this is not the case since it is computationally easy to do. I think that is ok. However, this explicitly leads to *data* overhead which could be used to have more accurate classifier and then a smaller prediction sets. I wonder if even this was taken into account for the experiments. From the paper itself, this is not clear.

- The authors use $\tilde \alpha$ multiple times without defining it properly. $\tilde \alpha_y$ is defined but not $\tilde \alpha$ which appears in Theorem 2 in several important part. For example when authors claim to achieve smaller confidence set, the two confidence level seems different which make the interpretation of this result difficult.

- It is also difficult to appreciate the Theorem 2. It is written in a convoluted way by introducing $\sigma_y$ which seems useless in the expression since it simplifies by definition (it was artificially introduced by multiplying and dividing by the denominator appearing in the definition of $\sigma_y$). Several reviewers mentioned this as well. We have

$$ \sum_{y \in \mathcal{Y}} \mathbb{P}\_{ X_{n+1} } \bigg[ V(X_{n+1}, y)  \leq \hat Q_{1 - \tilde \alpha}^{class}(y), r_f(X_{n+1}, y) \leq \hat k(y) \bigg] \leq   \sum_{y \in \mathcal{Y}} \mathbb{P}\_{ X_{n+1} } \bigg[ V(X_{n+1}, y)  \leq \hat Q_{1 - \alpha}^{class}(y) \bigg]  \text{ (Equation 12 ) }$$

(Eq. 12) is the assumption of theorem 2, is *equivalent* to the conclusion about the size of the set. What do we gain here?

- This lack of clarity is also in the proof of theorem 1 with statements as "P{Lemma 1 holds} + P{Lemma 1 not holds} ..."

The paper seems to tackle an important question but the contributions are unclear as of now. More iterations on the paper are needed. I cannot recommend this paper for ICLR, hence I suggest a reject.

**Justification For Why Not Higher Score:**

The paper presentation could be improved. Stronger results already exist in the literature, and the authors did not compare or properly their findings to them. Additionally, the impact of some hyperparameters is poorly explained. Overall, significant updates need to be made, and the paper should be reviewed again. It is currently below the acceptance threshold.

**Justification For Why Not Lower Score:**

N/A

---

### Decision · Program_Chairs · 2024-01-16

Reject